# Interplay of two transcription factors for recruitment of the chromatin remodeling complex modulates fungal nitrosative stress response

Yunqing Jian[1], Zunyong Liu[2], Haixia Wang[1], Yun Chen [1,2], Yanni Yin[1,2], Youfu Zhao [3] & Zhonghua Ma [1,2 ✉]

Nitric oxide (NO) is a diffusible signaling molecule that modulates animal and plant immune responses. In addition, reactive nitrogen species derived from NO can display antimicrobial activities by reacting with microbial cellular components, leading to nitrosative stress (NS) in pathogens. Here, we identify FgAreB as a regulator of the NS response in *Fusarium graminearum*, a fungal pathogen of cereal crops. FgAreB serves as a pioneer transcription factor for recruitment of the chromatin-remodeling complex SWI/SNF at the promoters of genes involved in the NS response, thus promoting their transcription. FgAreB plays important roles in fungal infection and growth. Furthermore, we show that a transcription repressor (Fglxr1) competes with the SWI/SNF complex for FgAreB binding, and negatively regulates the NS response. NS, in turn, promotes the degradation of Fglxr1, thus enhancing the recruitment of the SWI/SNF complex by FgAreB.

[1] State Key Laboratory of Rice Biology, Zhejiang University, Hangzhou, China. [2] Key Laboratory of Molecular Biology of Crop Pathogens and Insects, Institute of Biotechnology, Zhejiang University, Hangzhou, China. [3] Department of Crop Sciences, University of Illinois at Urbana-Champaign, Urbana, IL, USA. ✉email: zhma@zju.edu.cn

Nitric oxide (NO) is a highly diffusible and reactive signaling molecule in various biological systems, including microorganisms and plants. Previous reports revealed that NO plays a critical role in the host immunity of plants and animals[1–3]. Moreover, reactive nitrogen species (RNS) derived from NO and superoxide may possess antimicrobial activities and could directly react with various cellular components, including proteins, lipids, nucleic acids, thiols, and antioxidants[4–8]. Increasing knowledge from diverse systems indicates that hosts perceiving pathogens can provoke RNS, leading to nitrosative stress (NS)[4,7,9]. Many pathogens have thus developed various strategies to overcome NS during host-pathogen interactions[10,11].

In pathogenic bacteria, several regulators have been identified to modulate transcription of NS response genes[6,12]. NorR was first identified to sense NO via its mononuclear non-heme iron center and regulates the expression of *norVW* genes encoding nitric oxide detoxifying enzymes for NO detoxification[12]. NsrR is another regulator that senses NO through its [4Fe-4S] or [2Fe-2S] iron-sulfur cluster and modulates the expression of sixty NS response genes[10,12]. In addition, transcription regulators (TFs) such as Fnr, SoxR, and IscR also play important roles in NS response[12]. In the pathogenic fungus *Candida albicans*, the Cta4 TF has been reported to be involved in NS response[13]; however, the underlying regulatory mechanisms remain largely vague in this fungus, as well as in many others[10].

The ATP-dependent chromatin remodeling complex SWI/SNF is a junction where extracellular signaling pathways meet the chromatin[14,15]. Due to the low abundance of the SWI/SNF complex within cells and its non-sequence-specificity interaction with nucleosomal DNA, the complex needs to be "guided" to specific genomic regions by its co-operator(s)[16,17]. Chromatin-based epigenomic assays have shown that pioneer TFs may be able to recruit chromatin structure-related complexes to target regions in plants and animals[18–20]. In general, pioneer TFs have a specialized role in binding closed regions of chromatin and then initiating the subsequent opening of these regions. Thus, pioneer TFs have been considered as key factors in gene regulation with critical roles in developmental decisions, including organ biogenesis, tissue development, and cellular differentiation[21,22]. However, little is known to date as to how the SWI/SNF complex and pioneer TFs are involved in regulating gene transcription under environmental stress conditions, including NS in pathogenic fungi.

Here, we show that NO burst in wheat tissue inhibits the growth of *Fusarium graminearum* (named *Fg* hereafter), the major causal agent of Fusarium head blight (FHB), which is a devastating disease of cereal crops worldwide[23,24]. Our results demonstrate that FgAreB serves as a pioneer TF to recruit the SWI/SNF complex to the promoters of NS response genes. In addition, FgIxr1, a transcription repressor, prevents the interaction of the SWI/SNF complex with FgAreB. Under NS, degradation of FgIxr1 promotes the recruitment of the SWI/SNF complex by FgAreB at the promoters of NS response genes, leading to the high level of expression of these genes. These results indicate that the interplay of the two TFs plays a major role in the response of *Fg* to NS.

## Results

**Fg infection provokes NO burst in plant tissues.** In order to understand the role of NO in host defense against pathogen invasion, we first determined NO production in wheat coleoptile and corn stigma after *Fg* infection using NO-sensitive fluorescent probe DAF-FM DA (diaminofluorescein-FM diacetate)[26,27]. As shown in Fig. 1a, abundant NO was observed in infected wheat coleoptile and corn stigma cells at 24 h post-inoculation (hpi) with

*Fg* conidia or mycelia. When treated wheat coleoptile with chitin, NO production was visible in coleoptiles at 2 h post-treatment (hpt), indicating that the fungal cell wall component chitin induces NO production (Fig. 1a). When treated with the NO scavenger 2-(4-carboxyphenyl)-4, 4, 5, 5-tetramethylimidazoline-1-oxyl-3-oxide (cPTIO) at 100 μM, NO accumulation was abolished in wheat coleoptiles and corn stigma inoculated with *Fg* (Fig. 1a). These results indicate that *Fg* infection provokes NO burst in its host tissues.

To determine the effect of NO burst on fungal growth and development, we treated the wild-type strain PH-1 with a NO donor, sodium nitroprusside (SNP). We conducted this experiment on the minimal medium without nitrogen (MM-N) since nitrogen induces NO production in fungi, including *Aspergillus nidulans*, *Ganoderma lucidum*, and *Magnaporthe oryzae*[28–30] and *Fg* (Supplementary Fig. 1a). SNP at 10 mM, but not at 2 mM, induced NO accumulation dramatically in *Fg* hyphae (Fig. 1b), and inhibited hyphal growth (Fig. 1c), conidiation, and perithecium production (Supplementary Fig. 1b, c). In addition, SNP at 10 mM inhibited *Fg* hyphal growth and fungal biomass in nitrogen-containing media (PDA, MM, PDB, and YEPD) (Supplementary Fig. 1d, e). These results indicate that NO accumulation induced by 10 mM SNP inhibits *Fg* growth and development.

**FgAreB is a regulator for NS response in Fg.** Previous studies have documented that several inducible NO detoxifying enzymes are responsible for NS response in fungi[8,31,32]. In order to identify NS response regulator(s) in *Fg*, we screened more than 1500 targeted *Fg* gene deletion mutants for their sensitivity to SNP and identified one mutant at the locus of *FGSG_16452*, which showed dramatically increased sensitivity to 2 mM SNP and various nitrogen sources as compared to the wild-type PH-1 (Fig. 1d, Supplementary Fig. 2a, and Supplementary Data 1). The *FGSG_16452* locus is predicted to encode a protein with 470 amino acids (Fig. 1e), which shares 44.4% identity with AreB of *A. nidulans* (Supplementary Fig. 2b). Thus, FGSG_16452 was designated as FgAreB thereafter. FgAreB contains a typical zinc-finger GATA-binding domain (Fig. 1e), which is conserved in many other fungi including *Neurospora crassa* and *M. oryzae*[33,34] (Supplementary Fig. 2b). *Fg* contains five typical GATA TFs (FgAreB, FgAreA, FgSreA, FgWC1, and FgWC2). A previous study showed that mutations of FgWC1 (FGSG_07914) and FgWC2 (FGSG_00710) resulted in no phenotypic changes[35,36]. We, therefore, tested FgAreA and FgSreA mutants and found that both of the mutants showed no sensitivity to 2 mM SNP (Supplementary Fig. 2c), indicating FgAreB may be the only GATA-type TF responding to NS in *Fg*.

To confirm the role of the GATA-binding domain in FgAreB, we firstly constructed the FgAreB deletion mutant (ΔFgAreB) (Supplementary Fig. 3a–c). The mutant was then either complemented with the wild-type FgAreB or a truncated FgAreB without the GATA-binding domain fused with GFP under the control of its native promoter (Supplementary Fig. 3d). The resulting strains were named ΔFgAreB-C and ΔFgAreB-C$^{\Delta GATA}$, respectively. Phenotypic assays showed that the defects of mycelial growth, conidiation, and perithecium formation were complemented in ΔFgAreB-C, but not in ΔFgAreB-C$^{\Delta GATA}$ (Supplementary Fig. 3e–g); while GFP was observed in the nucleus in both ΔFgAreB-C and ΔFgAreB-C$^{\Delta GATA}$ (Supplementary Fig. 3h). In many other GATA-type TFs including FgSreA, the GATA domain overlaps with the nuclear localization signal (NLS) domain; however, the NLS domain is located in the C-terminus of FgAreB (Supplementary Fig. 3i). These results indicate that the GATA domain is essential for FgAreB function, but is dispensable for FgAreB localization.

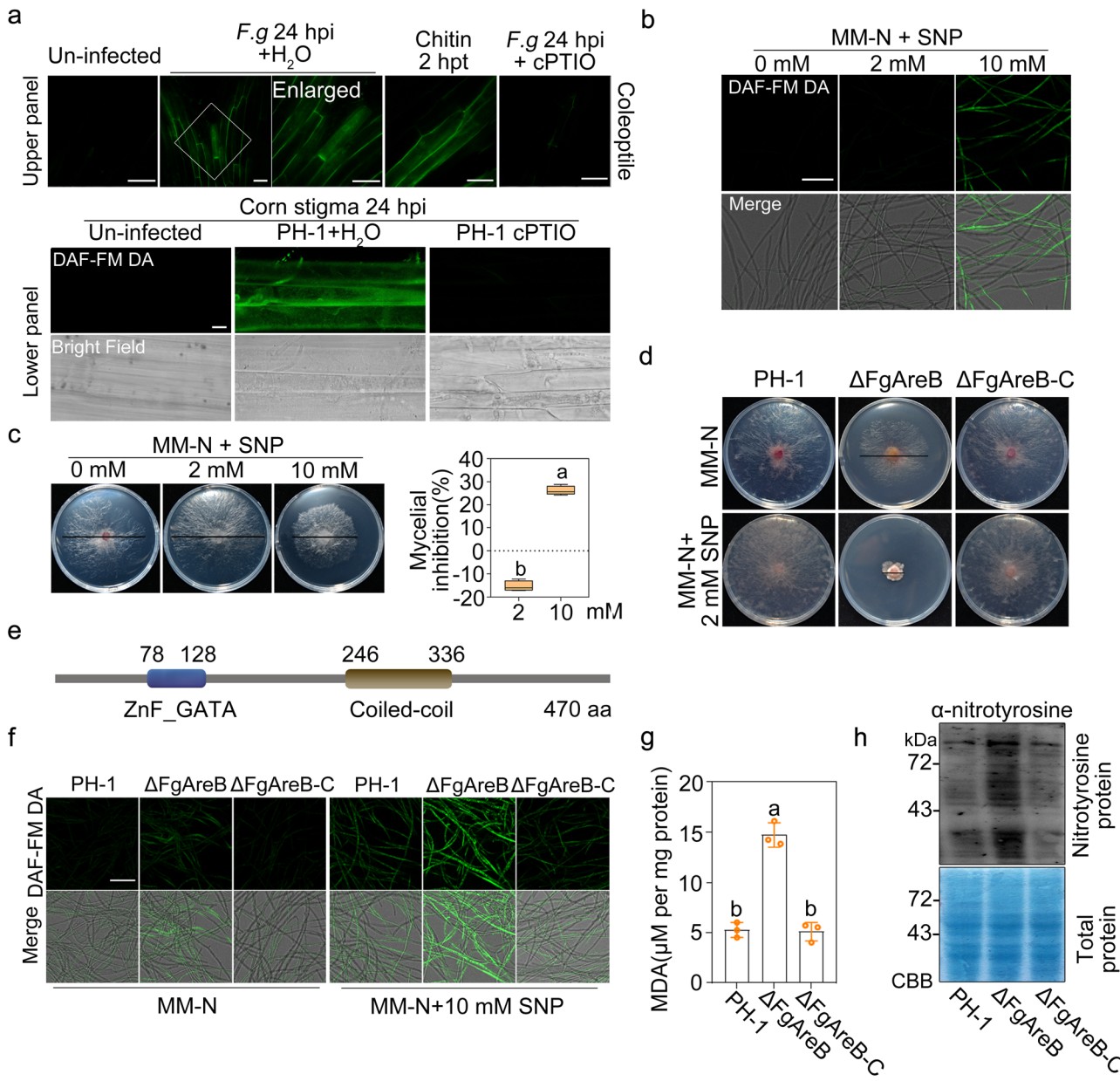

To further verify the association of FgAreB with NS response, we determined NO accumulation in ΔFgAreB. Microscopy examination showed that NO accumulation in ΔFgAreB was much higher than that of the wild-type PH-1 on MM-N with or without SNP and various nitrogen (Fig. 1f and Supplementary Fig. 1a). Phenotypic assays showed that ΔFgAreB was severely impaired in conidiation (Supplementary Fig. 3f) and was unable to produce perithecium (Supplementary Fig. 3g), which are similar to the wild-type PH-1 treated with 10 mM SNP (Supplementary Fig. 1b and c). In addition, ΔFgAreB contained higher levels of malondialdehyde (Fig. 1g) and nitrotyrosine (Fig. 1h) than those of the wild-type PH-1, both are markers for NS[28]. These results indicate that FgAreB is involved in modulating NS response in *Fg*.

**FgAreB regulates transcription of NS response genes in *Fg*.** Previous studies showed that five groups of NS response genes are responsible for detoxifying NO in fungi, i.e., flavohaemoglobins (FHB), S-nitrosoglutathione reductase (GSNOR), P450 nitric

oxide reductases (NOR), porphobilinogen deaminase (HEMC), and nitrosothionein[8,31,32]. BLASTp assay showed that *Fg* harbors five putative NS response genes, e.g., *FgFHB1* (FGSG_00765), *FgFHB2* (FGSG_04458), *FgGSNOR* (FGSG_10200), *FgNOR* (FGSG_11585), and *FgHEMC* (FGSG_08977). Except for *FgHEMC*, expression of four genes (*FgFHB1*, *FgFHB2*, *FgGSNOR*, and *FgNOR*) was induced by 10 mM SNP in the wild type, but not in ΔFgAreB (Fig. 2a), indicating that FgAreB modulates transcription of these four NS response genes.

To elucidate whether these FgAreB-dependent NS response genes are genetically involved in relieving NS, single, double, and triple mutants of these genes were constructed (Supplementary Fig. 4a). The resulting mutants were tested for sensitivity to SNP on MM-N medium. We found that two triple mutants ΔFgNSR1 (ΔFgFHB1-FHB2-GSNOR) and ΔFgNSR2 (ΔFgFHB1-GSNOR-NOR) exhibited significantly increased sensitivity to 2 mM SNP and various nitrogen sources (Fig. 2b; Supplementary Fig. 2a), while single and double mutants displayed slightly increased sensitivity to 2 mM SNP (Supplementary Fig. 4b). Furthermore, the triple mutants also accumulated higher levels of NO (Fig. 2c),

**Fig. 1 FgAreB mediates the response of *F. graminearum* to nitrosative stress. a** *F. graminearum* provoked NO in wheat coleoptiles (upper panel) and corn stigma (lower panel). Wheat coleoptiles were challenged with conidia of *F. graminearum* or 10 mg/ml chitin for the indicated time and then stained by DAF-FM DA with or without the NO scavenger cPTIO at 100 μM. Corn stigma was inoculated with mycelial plugs of *F. graminearum* at 24 hpi, and then stained by DAF-FM DA with or without cPTIO. Bars: 50 μm (upper panel), 10 μm (lower panel). The experiment was repeated three times independently with similar results. **b** SNP induced NO production in *F. graminearum*. Each strain was cultured in YEPD for 16 h and transferred to MM-N with 0, 2, or 10 mM SNP for 4 h. The resulting mycelia of each strain were stained with DAF-FM DA. NO fluorescence was detected by confocal microscopy. Bars: 50 μm. The experiment was repeated three times independently with similar results. **c** SNP inhibited the mycelial growth of *F. graminearum*. A 5-mm mycelial plug of PH-1 was inoculated on each MM-N plate supplemented with 0, 2, or 10 mM SNP and incubated at 25 °C for three days. Black lines indicate colony diameters on plates (left panel). The percentage of mycelial inhibition was calculated for each treatment (right panel). Data are shown as box plots with the interquartile range as the upper and lower confines of the box, and the median as a solid line within the box. Different letters indicate statistically significant differences according to the two-tailed Student's *t*-test ($p < 0.05$). **d** ΔFgAreB exhibited increased sensitivity to SNP. A 5-mm mycelial plug of the mutant was inoculated on each MM-N plate supplemented with 0 or 2 mM SNP and incubated at 25 °C for seven days. Black lines indicate colony diameters on the plates. **e** FgAreB contains a typical GATA zinc finger domain (ZnF-GATA) and a coiled-coil domain according to the SMART protein database (http://smart.embl-heidelberg.de) and NBCI protein database (https://blast.ncbi.nlm.nih.gov/Blast.cgi). **f** ΔFgAreB produced more NO as compared with the wild-type PH-1. Each strain was cultured in YEPD for 16 h and then transferred to MM-N with or without 10 mM SNP for 4 h. The resulting mycelia of each strain were stained with DAF-FM DA. Fluorescence was detected by confocal microscopy. Bars: 50 μm. The experiment was repeated three times independently with similar results. **g** ΔFgAreB showed increased malondialdehyde (MDA) under NS condition. MDA was determined for PH-1, ΔFgAreB, and ΔFgAreB-C (ΔFgAreB::FgAreB-GFP) after each strain was cultured in YEPD for 16 h and then transferred to MM-N with 10 mM SNP for 4 h. Data presented are the mean ± standard errors from three biological replicates ($n = 3$). Different letters represent statistically significant differences according to the one-way ANOVA test ($p < 0.05$). **h** Nitrotyrosine level was elevated in ΔFgAreB under NS condition. Proteins were extracted from each strain cultured in YEPD for 16 h and then transferred to MM-N with 10 mM SNP for 4 h. Nitrotyrosine was detected by western blot with anti-nitrotyrosine polyclonal antibody (upper panel). Protein loading control is shown by coomassie blue staining (lower panel). The experiment was repeated three times independently with similar results.

nitrotyrosine (Fig. 2d), and malondialdehyde (Fig. 2e) as compared with those in PH-1. These results indicate that *FgFHB1*, *FgFHB2*, *FgGSNOR*, and *FgNOR* are involved in relieving NS in *Fg*.

The GATA TFs bind the sequence-specific *cis*-element containing six nucleotides HGATAR (H=A, C, T, and R=A, G) through their zinc-finger GATA-binding domain[37,38]. Bioinformatics analysis showed that the promoters of these four NS response genes (*FgFHB1*, *FgFHB2*, *FgGSNOR*, and *FgNOR*), but not *FgHEMC*, contain putative FgAreB-binding *cis*-element (Fig. 2f). Electrophoretic mobility shift assay (EMSA) using the purified His-tagged GATA binding domain of FgAreB, FgAreA, and FgSreA (named FgAreB^GATA-His, FgAreA^GATA-His, and FgSreA^GATA-His, respectively (Supplementary Fig. 4c) showed that FgAreB^GATA-His, but FgAreA^GATA and FgSreA^GATA, was able to bind to the promoters of *FgFHB1*, *FgFHB2*, *FgGSNOR*, and *FgNOR*, but not that of *FgHEMC* (Fig. 2f; Supplementary Fig. 4d). Moreover, competition EMSA assay showed that only the wild type *cis*-element, but not the mutated *cis*-element, inhibited band shift (Fig. 2f), indicating that the FgAreB binds specifically to the *cis*-element in vitro.

Chromatin immunoprecipitation-quantitative PCR (ChIP-qPCR) assay showed that FgAreB-GFP was significantly enriched at the promoters of four NS response genes, but not at that of *FgHEMC* in the complementary strain (Fig. 2g). FgAreA-GFP and FgSreA-GFP were also unable to enrich at the *FgGSNOR* promoter (Supplementary Fig. 4e). These results suggest that FgAreB binds to the promoters of NS response genes containing the *cis*-element HGATAR in *Fg*.

In mammals, the GATA TFs have been identified as pioneer TFs that bind directly to nucleosomal wrapped DNA[19]. To further explore whether FgAreB enriches the promoter region wrapped nucleosomal DNA in the absence of NS, we performed a micrococcal nuclease (MNase)-qPCR assay for the wild-type strain without SNP treatment. Results showed that the promoter region containing the FgAreB binding *cis*-element ($-380 \sim -220$) of *FgGSNOR* wrapped around one nucleosome (Fig. 2h), indicating FgAreB was able to enrich at the nucleosomal wrapped DNA region. This is consistent with the results that FgAreB-GFP was still able to enrich at the promoters of four NS response genes

in the wild type without SNP treatment in the ChIP-qPCR assays (Fig. 2g). These results suggest that FgAreB is able to enrich at nucleosomal wrapped DNA region.

**FgAreB associates with the SWI/SNF complex via direct interaction with FgSwp73 to mediate *Fg* response to NS.** To further elucidate the regulatory mechanism of FgAreB in response to NS, we conducted yeast two-hybrid (Y2H) screening to identify potential FgAreB-interaction proteins using *Fg* cDNA library as prey and FgAreB as a bait. A total of 51 putative FgAreB interacting proteins were recovered in the screening (Supplementary Data 2). Among them, FgSwp73 (Fig. 3a), a homolog of *S. cerevisiae* Swp73, is a core component of the SWI/SNF chromatin remodeling complex[39–41]. BLASTp search using subunits of the yeast SWI/SNF complex as queries identified ten counterparts of the SWI/SNF subunits in *Fg* (Supplementary Table 1). Y2H assays showed that FgAreB interacted only with FgSwp73, but not with other subunits of the complex (Fig. 3a; Supplementary Fig. 5a); whereas FgSwp73 interacted with FgSnf2, FgSnf5, FgSwi3, FgSwi1, and FgArp7 (Fig. 3b; Supplementary Fig. 5b). These results indicate that FgAreB is most likely associated with the SWI/SNF complex via interaction with FgSwp73. Moreover, Y2H assays using a series of truncated FgAreB and GST-pull-down assay demonstrated that the GATA domain of FgAreB is critical for its interaction with FgSwp73 (Fig. 3c; Supplementary Fig. 5c, d). In addition, protein subcellular localization experiments revealed that FgAreB co-localized with the core SWI/SNF subunits FgSwp73, FgSnf2, and FgSnf5 into the nucleus (Fig. 3d; Supplementary Fig. 5e, f). Co-Immunoprecipitation (Co-IP) assays further confirmed that FgAreB interacted with FgSwp73, FgSnf2, and FgSnf5 in vivo, and their interactions were enhanced by SNP treatment (Fig. 3e; Supplementary Fig. 5g, h). These results indicate that FgAreB associates with the SWI/SNF complex via direct interaction with FgSwp73.

To explore the genetic association of FgAreB and the SWI/SNF complex, we tried to construct deletion mutants of each SWI/SNF subunit. Among the 10 subunits, six (FgSwi3, FgSwi1, FgArp9, FgSwp82, FgSnf5, and FgTaf14) were deleted successfully (Supplementary Fig. 6a), whereas the remaining four subunits

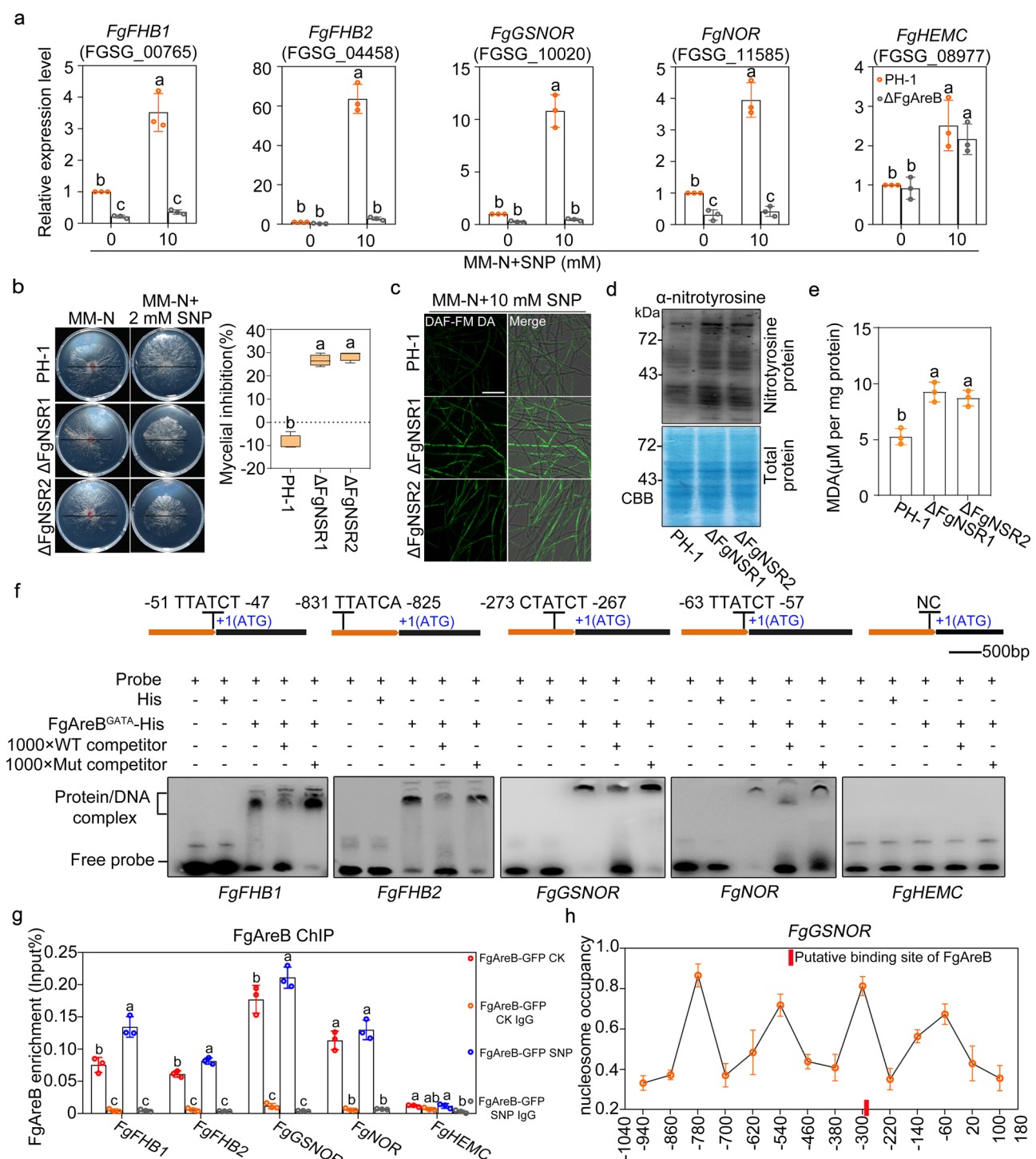

(FgSwp73, FgArp7, FgSnf6, and FgSnf2) were not, indicating these four subunits might be essential for *Fg* growth although the Swp73 orthologs were not essential in *S. cerevisiae*[39] and *N. crassa*[42]. Phenotypic assays showed that the FgSnf5 mutant exhibited dramatically increased sensitivity to 2 mM SNP and various nitrogen sources (Fig. 3f and Supplementary Fig. 2a), while the FgArp9 mutant was also sensitive to SNP (Supplementary Fig. 6b). Further characterization of the FgSnf5 mutant showed that the mutant accumulated higher levels of NO (Fig. 3g and Supplementary Fig. 1a), malondialdehyde (Fig. 3h), and nitrotyrosine (Fig. 3i), as compared with those in PH-1. In addition, the FgSnf5 mutant was unable to produce perithecium

and displayed a defect in conidiation (Supplementary Fig. 6c and d). The four NS response genes were also dramatically downregulated in the mutant (Fig. 3j). ChIP-qPCR assay showed that SNP treatment promoted the enrichment of FgSnf5-GFP at the promoters of these genes (Fig. 4a). These results indicate that the SWI/SNF complex is involved in regulating NS response at least partially via modulating the transcription of NS response genes.

## FgAreB recruits the SWI/SNF complex to mediate chromatin accessibility

Several previous studies have documented that the SWI/SNF complex needs to be "guided" to target genes by specific

**Fig. 2 FgAreB regulates nitrosative stress response genes in *F. graminearum*. a** FgAreB upregulated expression of *FgFHB1*, *FgFHB2*, *FgGSNOR*, and *FgNOR*, but not *FgHEMC* after SNP treatment. Each strain was cultured in YEPD for 16 h and then transferred to MM-N without or with 10 mM SNP for 4 h. The *FgACTIN* gene was used as an internal control. Data presented are the mean ± standard errors from three biological replicates ($n = 3$). Different letters represent statistically significant differences according to the one-way ANOVA test ($p < 0.05$). **b** The ΔFgNSR1 (ΔFgFHB1/FHB2/GSNOR) and ΔFgNSR2 (ΔFgFHB1/GSNOR/NOR) triple mutants showed increased sensitivity to SNP. A 5-mm mycelial plug of each strain was incubated for three days on MM-N plate supplemented with or without 2 mM SNP for three days (left panel). Black lines indicate colony diameters. The percentage of mycelial inhibition was calculated (right panel). Data are shown as box plots with the interquartile range as the upper and lower confines of the box, and the median as a solid line within the box. Different letters indicate statistically significant differences according to the one-way ANOVA test ($p < 0.05$). **c** ΔFgNSR1 and ΔFgNSR2 accumulated more NO under SNP treatment. NO in fungal hypha was stained with DAF-FM DA and the fluorescence was observed using confocal microscopy. Bars: 50 μm. The experiment was repeated three times independently with similar results. **d** ΔFgNSR1 and ΔFgNSR2 exhibited increased nitrotyrosine levels under SNP treatment. Nitrotyrosine was detected by anti-nitrotyrosine polyclonal antibody (upper panel). Protein loading control is shown by coomassie blue staining (CBB) (lower panel). The experiment was repeated three times independently with similar results. **e** ΔFgNSR1 and ΔFgNSR2 showed increased malondialdehyde (MDA) levels under SNP treatment. Data presented are the mean ± standard errors from three biological replicates ($n = 3$). Different letters represent statistically significant differences according to the one-way ANOVA test ($p < 0.05$). **f** Electrophoretic mobility shift assay (EMSA) shows that FgAreB binds the GATA *cis*-element (HGATAR) in the promoters of *FgFHB1*, *FgFHB2*, *FgGSNOR*, *FgNOR*, but not *FgHEMC*. The schematic diagram shows the predicted binding *cis*-elements of FgAreB in the promoter region of NS response genes (upper panel). NC: No *cis*-element. The experiment was repeated twice independently with similar results. **g** ChIP-qPCR assay showed the enrichment of FgAreB at promoters of the *FgFHB1*, *FgFHB2*, *FgGSNOR*, and *FgNOR* genes without or with 10 mM SNP treatment. The input-DNA and ChIP-DNA samples were quantified by quantitative PCR assays with corresponding primer pairs (Supplementary Data 11). Secondary antibody rabbit IgG was used as a control. ChIP signals are shown as the percentages of input. Data presented are the mean ± standard errors from three biological replicates ($n = 3$). Different letters represent statistically significant differences according to the one-way ANOVA test ($p < 0.05$). **h** Nucleosome occupancy at *FgGSNOR* promoter in wild type PH-1 without SNP treatment. Nucleosome occupancy in PH-1 growth in MM-N was determined by MNase treatment followed by qPCR analysis. Four nucleosomes to be wrapped were detected for the 1-kb *FgGSNOR* promoter region which contains the putative FgAreB binding site (Red box). Data presented are the mean ± standard errors from three biological replicates ($n = 3$).

factors due to its non-sequence-specific interaction with nucleosomal DNA[17,43–46]. Since the association of FgAreB with the SWI/SNF complex was enhanced by SNP treatment (Fig. 3e; Supplementary Fig. 5g, h), we hypothesized that FgAreB might recruit this complex to the promoters of NS response genes and activates their transcription. To test this hypothesis, we performed ChIP-qPCR analysis to measure the occupancy pattern of FgSnf5 in the promoters of NS response genes in the FgAreB deletion mutant (ΔFgAreB::FgSnf5-GFP) and vice versa. SNP treatment promoted the enrichment of FgSnf5-GFP at NS response gene promoters significantly in the wild-type strain (PH-1::FgSnf5-GFP), but not in the ΔFgAreB stain (ΔFgAreB:: FgSnf5-GFP), indicating that the enrichment of FgSnf5 largely relies on FgAreB (Fig. 4a). In contrast, the enrichment of FgAreB-GFP at the promoters of NS response genes was independent of FgSnf5 and SNP treatment (Fig. 4b; Fig. 2g). Furthermore, ChIP-qPCR and MNase-qPCR assays showed that nucleosome occupancy at the FgAreB putative binding site of *FgGSNOR* promoter in ΔFgAreB and ΔFgSnf5 was higher than that in PH-1 with or without SNP treatment (Fig. 4c, d), indicating that nucleosomes at the *FgGSNOR* promoter were unable to exit effectively in the absence of FgAreB or FgSnf5. In addition, loss of *FgAreB* or *FgSnf5* gene did not affect each other's transcription (Supplementary Fig. 7a–b). These results indicate that FgAreB may act as a pioneer TF that recruits the SWI/SNF complex to the promoters of NS response genes.

Considering both ΔFgAreB and ΔFgSnf5 exhibited much more severe growth defect than the triple mutants ΔFgNSR1 and ΔFgNSR2, it is reasonable to believe that in addition to NS response, FgAreB and FgSnf5 might be involved in regulating other functions in *F. graminearum*. RNA-seq data for ΔFgAreB and ΔFgSnf5 growth in a conventional medium PDB revealed that FgAreB and FgSnf5 co-activated 657 genes and co-suppressed 1431 genes (Fig. 4e; Supplementary Fig. 8a; Supplementary Data 3–5; Supplementary Data 6–8). Consistent with the RT-qPCR results (Fig. 2a and Fig. 3j), NS response genes *FgFHB1*, *FgFHB2*, and *FgNOR* were dramatically downregulated in both ΔFgSnf5 and ΔFgAreB mutants (Supplementary Table 2). Enrichment analysis of Gene Ontology (GO) of downregulated

genes indicates FgAreB and FgSnf5 are involved in some key metabolic processes, including ADP, purine ribonucleoside and nucleoside, pyridine nucleoside, monosaccharide, which may partially explain the severe growth defect of ΔFgAreB and ΔFgSnf5 (Supplementary Fig. 8b; Supplementary Data 9). On the other hand, GO related to condensed chromosome kinetochore, DNA repair, DNA metabolic, cellular response to DNA damage stimulus, chromosome, DNA replication was significantly enriched in upregulated genes in both ΔFgAreB and ΔFgSnf5, which may be due to changed chromosome accessibility from deletion of *FgAreB* and *FgSnf5* (Supplementary Fig. 8c; Supplementary Data 10).

To further verify the role of FgSnf5 and FgAreB in regulating chromatin accessibility, ATAC-seq (Assay for Transposase Accessible Chromatin, combined with high throughput sequencing) was performed to profile chromatin landscape (Supplementary Fig. 9a–d). As shown in Fig. 4f, deletion of *FgAreB* or *FgSnf5* led to 2050 or 1263 inaccessible genome peaks, respectively, as compared to the wild type. Intriguingly, 1051 peaks were co-regulated by FgAreB and FgSnf5, indicating that more than 83% of the inaccessible peaks in ΔFgSnf5 were also inaccessible in ΔFgAreB (Fig. 4f, left panel; Supplementary Fig. 9e). On the other hand, ΔFgAreB and ΔFgSnf5 also showed 1494 and 2063 increased accessible peaks, respectively, as compared to the wild type (Fig. 4f, right panel). Among them, 802 increased accessible peaks were co-regulated by FgAreB and FgSnf5 (Fig. 4f, right panel), indicating that only 38.9% of the increased accessible peaks in ΔFgSnf5 were also accessible in ΔFgAreB. Consistently, the summit-centered heatmap also indicates that the peaks of chromatin inaccessibility in ΔFgAreB were highly overlapped with that of ΔFgSnf5 (Fig. 4g, h). Taken together, these results imply that FgAreB collaborates with FgSnf5 to mainly modulate chromatin accessibility, although they also play a role in regulating chromatin inaccessibility.

Notably, FgAreB itself regulates both chromatin accessibility (2050/3544 peaks) and inaccessibility (1494/3544 peaks), which is similar to other TFs reported previously. In human cells, the number of accessible loci modulated by TFs (such as the pioneer TF GATA3, the nuclear factors IB, and IX) is comparable to that

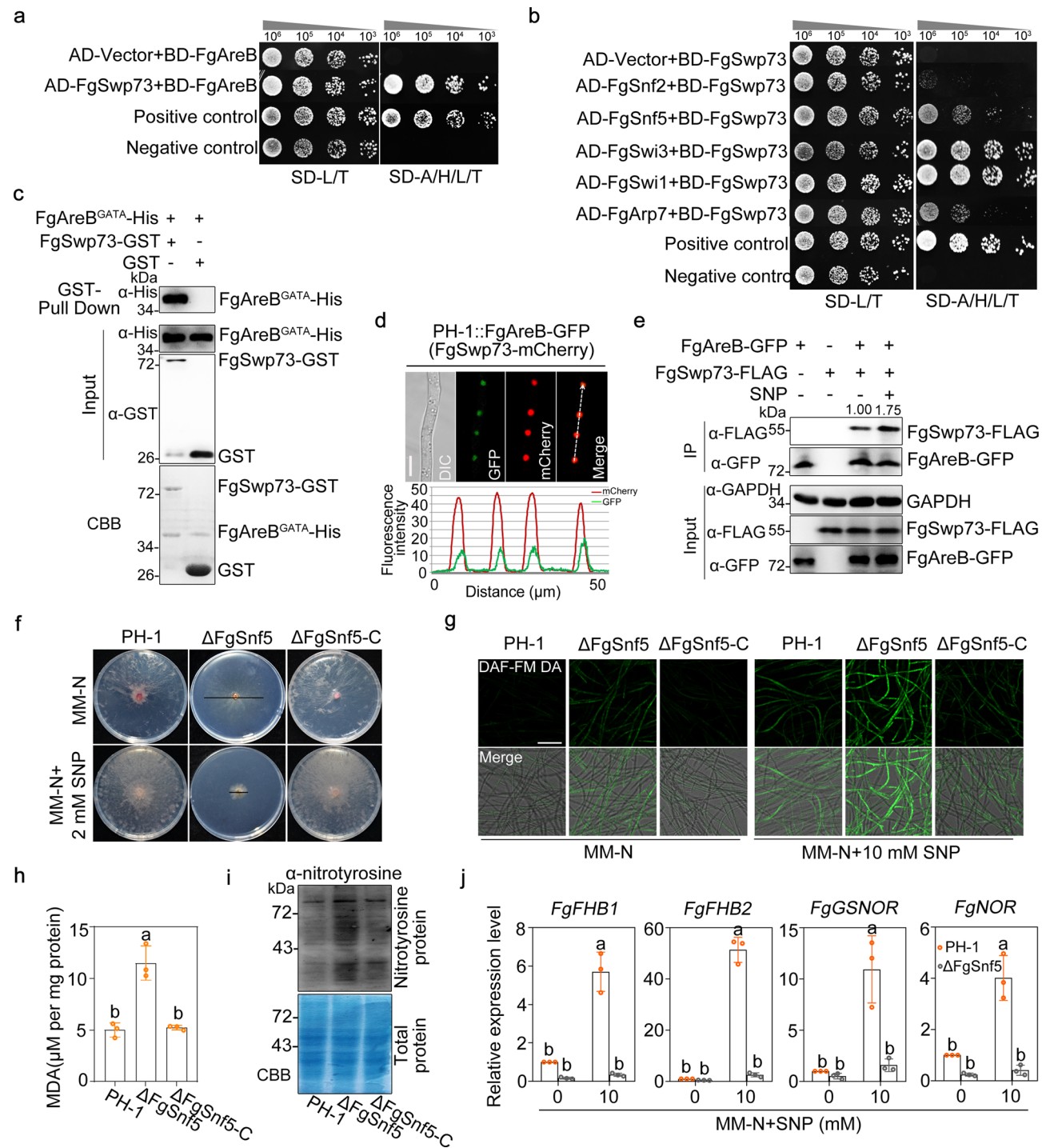

of inaccessible loci, indicating other factors may be involved in accessing these loci[47,48].

**FgIxr1 competitively interacts with FgAreB to prevent the interaction of FgAreB with the SWI/SNF complex.** In the cDNA library screening assay, another interactor of FgAreB, FgIxr1, is a homolog in *S. cerevisiae* which encodes a transcription repressor. Phenotypic assays revealed that the FgIxr1 mutant (ΔFgIxr1) exhibited increased resistance to 10 mM SNP as compared with the wild type (Fig. 5a, Supplementary Fig. 10a). Consistently, transcription of NS response genes was higher in ΔFgIxr1 in response to SNP treatment (Fig. 5b), indicating that

FgIxr1 might negatively affect NS response. The ChIP-qPCR assay showed that enrichment of FgIxr1-GFP at the *FgGSNOR* gene promoter decreased significantly after SNP treatment (Supplementary Fig. 10b). ΔFgIxr1 also exhibited decreased NO production, nitrotyrosine, and malondialdehyde (Supplementary Fig. 10c–e), further suggested that FgIxr1 negatively regulates NS response through repressing the expression of NS response genes.

To elucidate the transcriptional repression mechanism of NS response genes by FgIxr1, we first examined the localization of FgIxr1 and found that FgIxr1-mCherry co-localized with FgAreB-GFP into the nucleus under MM-N condition (Fig. 5c). The interaction of FgAreB and FgIxr1 was then confirmed by Y2H and Co-IP assays (Fig. 5d, e). Interestingly, the GATA domain of

**Fig. 3 FgAreB associates with the SWI/SNF complex to regulate transcription of NS response genes. a** FgAreB interacted with FgSwp73 in Y2H assay. Interactions were determined by monitoring the growth on synthetic defined (SD) medium lacking leucine (L), tryptophan (T), histidine (H), and adenine (A) (SD-L/T/H/A) of yeast cells bearing a pair of vectors as indicated. pGBKT7-53 and pGADT7 used as positive control. pGBKT7-Lam and pGADT7 used as negative control. The experiment was repeated three times independently with similar results. **b** Determination of the interactions of FgSwp73 with FgSnf2, FgSnf5, FgSwi3, FgSwi1, and FgArp7 in Y2H assays. The assays were performed as described in **a**. **c** The GATA-domain of FgAreB (FgAreB$^{GATA}$) interacted with FgSwp73 in vitro by GST pull-down assay. GST or FgSwp73-GST immobilized on GST beads was incubated with the FgAreB$^{GATA}$-His protein. The beads were washed and pelleted for immunoblotting with anti-His antibody. Immunoblotting with anti-His antibody and anti-GST antibody was used for the detection of input proteins. The input proteins were also detected by staining with coomassie brilliant blue (CBB). The experiment was repeated twice independently with similar results. **d** FgAreB-GFP and FgSwp73-mCherry co-localized into nucleus. The wild-type background strain bearing FgAreB-GFP and FgSwp73-mCherry constructs were grown in MM-N and then examined using confocal microscopy (upper panel). Bars: 10 μm. Co-localization of proteins was further evaluated by line-scan graph analysis (lower panel). The white dotted arrow indicates the analyzed area and the horizontal axis indicates the distance. The experiment was repeated twice independently with similar results. **e** The Association of FgAreB with FgSwp73 was determined by co-immunoprecipitation (Co-IP) assay. Proteins were extracted from the wild-type background strain containing FgAreB-GFP and FgSwp73-3×FLAG constructs or a single construct growth in liquid MM-N with or without SNP treatment, and were immunoprecipitated with anti-GFP agarose beads; immunoblotted with anti-FLAG antibody and anti-GFP antibody (upper panel). The lower panel shows the input samples. The experiment was repeated twice independently with similar results. **f** ΔFgSnf5 exhibited increased sensitivity to SNP. A 5-mm mycelial plug of each strain was inoculated on each MM-N plate with or without 2 mM SNP and incubated at 25 °C for seven days. Black lines indicate colony diameters. **g** ΔFgSnf5 showed increased NO production under SNP treatment. The mycelia of each strain cultured in YEPD for 16 h was transferred to MM-N with or without 10 mM SNP for 4 h and was stained with DAF-FM DA. Fluorescence was detected by confocal microscopy. Bars: 50 μm. The experiment was repeated three times independently with similar results. **h** ΔFgSnf5 exhibited increased malondialdehyde (MDA) under SNP treatment. Data presented are the mean ± standard errors from three biological replicates ($n = 3$). Different letters represent statistically significant differences according to the one-way ANOVA test ($p < 0.05$). **i** ΔFgSnf5 exhibited increased nitrotyrosine under SNP treatment. Nitrotyrosine was detected by western blot with anti-nitrotyrosine polyclonal antibody (upper panel). CBB shows protein loading control (lower panel). The experiment was repeated three times independently with similar results. **j** FgSnf5 was required for transcriptional upregulation of *FgFHB1*, *FgFHB2*, *FgGSNOR,* and *FgNOR* under SNP treatment. *FgACTIN* gene used as an internal control. Data presented are the mean ± standard errors from three biological replicates ($n = 3$). Different letters represent statistically significant differences according to the one-way ANOVA test ($p < 0.05$).

FgAreB was also critical for this interaction (Fig. 5d–f) as the truncated FgAreB without the GATA domain (FgAreB$^{\Delta GATA}$) significantly decreased its interaction with FgIxr1 (Fig. 5d, e). Considering that the GATA domain of FgAreB is critical for its interaction with FgSwp73 (Supplementary Fig. 5d), we speculated that FgIxr1 might compete with FgSwp73 in their interactions with FgAreB. Yeast two-hybrid and yeast three-hybrid assays between FgSwp73 and FgAreB fused or co-expressed with or without FgIxr1 showed that FgIxr1 interfered with the interaction between FgAreB and FgSwp73 (Fig. 5g; Supplementary Fig. 10f). Furthermore, in vitro pull-down assay of FgSwp73-GST and FgAreB$^{GATA}$-His in different concentrations of FgIxr1-His showed that the amounts of FgAreB on FgSwp73-resin were gradually decreased as the amount of FgIxr1 increased (Fig. 5h). In parallel, Co-IP assays also confirmed that the FgAreB-FgSwp73 interaction was enhanced in ΔFgIxr1 as compared with that in the wild type (Supplementary Fig. 10g). These results indicate that FgIxr1 might compete with FgSwp73 in their interactions with FgAreB.

To gain further insight into the mechanism of FgIxr1 in regulating transcription of NS response genes, we determined the transcription and translation levels of FgIxr1 under SNP treatment. The RT-qPCR assay showed that transcription of FgAreB, FgSnf5, and FgIxr1 did not change significantly by SNP treatment (Supplementary Fig. 10h). Whereas FgIxr1 protein was not stable and showed degradation with the extension of SNP treatment time or increasing SNP concentration (Fig. 5i; Supplementary Fig. 10i). In contrast, FgAreB and FgSnf5 proteins were quite stable under the same conditions (Supplementary Fig. 10j). Consistently, microscopy observation also showed that SNP treatment led to decreased fluorescence and protein intensity of FgIxr1-mCherry (Fig. 5j), and SNP plus the protease inhibitor MG132 treatment alleviated the degradation of FgIxr1 (Fig. 5i). As a control, the protein accumulation of FgIxr1 was not affected after treated with fungicide carbendazim, which targets fungal tubulin (Supplementary Fig. 10k). These data indicate that SNP treatment promotes FgIxr1 degradation.

As described above, *Fg* infection provokes NO burst in plant tissues. We inoculated the ΔFgIxr1::FgIxr1-mCherry strain on wheat leaves and examined the intensity of mCherry fluorescence and protein levels. Microcopy and western blot assay showed the fluorescence intensity and protein level of FgIxr1-mCherry in the hyphae at 12 hpi was obviously lower than that at 2 hpi (Fig. 5k). Taken together, our results indicate that NS promotes FgIxr1 degradation, thus enhancing the recruitment of the SWI/SNF complex by FgAreB and promoting transcription of NS response genes.

**FgAreB and FgSnf5 are required for virulence of *F. graminearum*.** In order to determine whether NS response of pathogenic fungi is important for virulence, we examined the virulence of ΔFgAreB, ΔFgSnf5, ΔFgNSR1, ΔFgNSR2, and ΔFgIxr1 on flowering wheat heads and wheat coleoptile. Both ΔFgAreB and ΔFgSnf5 were non-pathogenic on the wheat head and wheat coleoptile; whereas ΔFgIxr1, ΔFgNSR1, and ΔFgNSR2 all showed reduced virulence (Fig. 6a, b). It has been reported that nitric oxide derivatives can trigger ROS production in plant tissues[49]. Therefore, the accumulation of ROS in wheat coleoptiles was examined by staining with 3, 3'-diamino-benzidine (DAB). Results showed that higher levels of ROS accumulation were observed in wheat coleoptile cells infected with the ΔFgAreB, ΔFgSnf5, ΔFgNSR1, and ΔFgNSR2 mutants than those infected with PH-1, ΔFgIxr1, and complementary strains (Supplementary Fig. 11). Furthermore, since NS response genes of *Fg* are responsible for overcoming NS, the expression of these genes during fungal infection was dramatically higher in PH-1 and ΔFgIxr1 than those in ΔFgAreB and ΔFgSnf5 at 12 hpi (Fig. 6c).

Previous studies have shown that NO also activates the transcription of plant defense-related genes[50,51], we thus examined the expression of selected defense-related genes in wheat seedling leaves after inoculation. Expression of *TaPR1*, *TaPR5,* and *TaWRKY23* genes was much higher in wheat cells

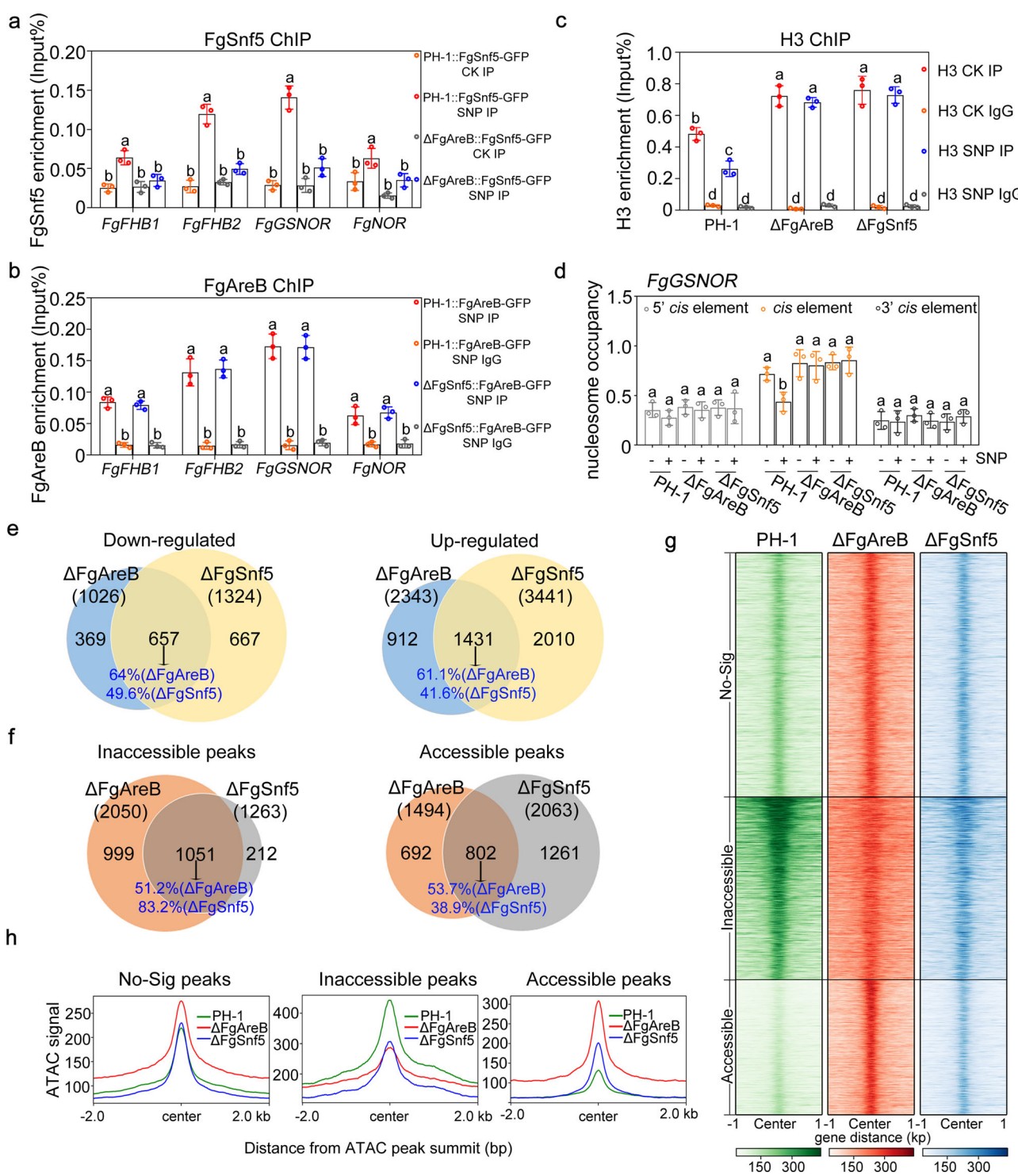

inoculated with ΔFgAreB, ΔFgSnf5, ΔFgNSR1, and ΔFgNSR2 mutants than those inoculated with PH-1 and ΔFgIxr1 (Fig. 6d). These results suggest that the ability to subverting NO burst in *Fg* may partially contribute to its virulence on the wheat host. Given that mycotoxin, DON is an important virulence factor of *Fg*[52,53] and the enzyme calonectrin oxygenase (Tri1) catalyzing the late step of DON biosynthesis mainly localizes to toxisomes in the mycotoxin induction medium (TBI)[54,55], we examined the toxisome formation using FgTri1-GFP as a marker. We found that 10 mM SNP treatment inhibited toxisome formation and

accumulation of FgTri1-GFP protein (Supplementary Fig. 12). We also examined the toxisome formation for the mutants. Both ΔFgAreB and ΔFgSnf5 were unable to form normally green spherical and crescent-shaped toxisomes in TBI medium and on wheat tissue, whereas PH-1 and the complementary strains clearly formed toxisomes (Fig. 6e). Consistently, the GFP signal of FgTri1-GFP in ΔFgAreB and ΔFgSnf5 was undetectable (Fig. 6f), and DON production in ΔFgAreB, ΔFgSnf5, ΔFgNSR1, and ΔFgNSR2 decreased significantly as compared with that in the wild type (Fig. 6g). Overall, these results suggest that increased

**Fig. 4 FgAreB recruits the SWI/SNF complex to regulate gene expression and chromatin accessibility. a** FgAreB was required for the enrichment of FgSnf5 at the promoters of NS response genes under SNP treatment. The DNA sample was extracted from each strain cultured in YEPD for 16 h and then incubated for 4 h in MM-N supplemented without or with 10 mM SNP. The input-DNA and ChIP-DNA samples were quantified by quantitative PCR assays with corresponding primer pairs (Supplementary Data 11). ChIP signals are shown as the percentages of input. Data presented are the mean ± standard errors from three biological replicates ($n = 3$). Different letters represent statistically significant differences according to the one-way ANOVA test ($p < 0.05$). **b** FgSnf5 was not required for enrichment of FgAreB at the promoters of NS response genes under SNP treatment. The input-DNA and ChIP-DNA samples were quantified by quantitative PCR assays with corresponding primer pairs (Supplementary Data 11). Secondary antibody rabbit IgG was used as a control. ChIP signals are shown as the percentage of input. Data presented are the mean ± standard errors from three biological replicates ($n = 3$). Different letters represent statistically significant differences according to the one-way ANOVA test ($p < 0.05$). **c** ΔFgAreB and ΔFgSnf5 exhibited increased enrichment of histone H3 at the *FgGSNOR* promoter as compared with the wild-type PH-1 under SNP treatment. ChIP-DNA and input-DNA samples were quantified by quantitative PCR assays with the primer at *FgGSNOR* promoter. Secondary antibody rabbit IgG used as a control. ChIP signals are shown as the percentage of input. Data presented are the mean ± standard errors from three biological replicates ($n = 3$). Different letters represent statistically significant differences according to the one-way ANOVA test ($p < 0.05$). **d** ΔFgAreB and ΔFgSnf5 showed increased nucleosome retention at the *FgGSNOR* promoter as compared with the wild-type PH-1 under SNP treatment. Nucleosome occupancy was determined via MNase treatment followed by qPCR analysis with three pair primers covering the FgAreB binding *cis*-element region (*cis*-element), the 5′ of the region of *cis*-element (5′ *cis*-element), and 3′ region of *cis*-element (3′ *cis*-element). Data presented are the mean ± standard errors from three biological replicates ($n = 3$). Different letters represent statistically significant differences according to the one-way ANOVA test ($p < 0.05$). **e** Venn diagrams showed numbers of genes regulated by FgAreB and FgSnf5 in *F. graminearum* by comparing ΔFgAreB and ΔFgSnf5 mutants with the wild type using RNA-seq. The number of common genes was also indicated (middle). Supplementary Data 3-8 list differentially expressed genes. RNA samples for sequencing were prepared from the mycelia cultured in PDB medium. **f** FgAreB coordinated with FgSnf5 to regulate chromatin accessibility in *F. graminearum*. Venn diagrams show common inaccessible peaks (left panel) and accessible peaks (right panel) in ΔFgAreB and ΔFgSnf5 determined by ATAC-seq assay. **g** ATAC-seq summit-centered heatmap of ATAC-seq signals in the wild-type PH-1, ΔFgAreB, and ΔFgSnf5 in the same regions. The ATAC-seq peaks are sorted in: no-significant (No-Sig), accessible, and inaccessible groups for ΔFgAreB, and ΔFgSnf5 based on the background of the signals in PH-1. **h** Average diagram of total ATAC-seq signals of PH-1, ΔFgAreB, and ΔFgSnf5 in the same regions based on the classification in **g**.

sensitivity to NS and decreased DON production might contribute to the attenuated virulence of ΔFgAreB, ΔFgSnf5, ΔFgNSR1, and ΔFgNSR2 on wheat.

## Discussion

The GATA TFs are highly conserved and play important roles in regulating transcription of genes involved in development and response to environmental changes in eukaryotes[56]. In fungi, the GATA TFs serve as key regulators in nitrogen metabolism, light perception, siderophore biosynthesis, mating-type switching, and secondary metabolism[57,58]. In *N. crassa*, a GATA TF WC-1 recruits the SWI/SNF complex to the promoter of *frq* (frequency, a circadian pacemaker gene) to aid in remodeling the nucleosomes, and subsequently initiate a circadian cycle of *frq* transcription[42]. In vertebrates, several GATA TFs, including FoxA1, GATA3, and GATA4, possess a hallmark feature of pioneer TFs that can bind to nucleosomal DNA and induce chromatin accessibility[38,59]. In this study, we demonstrated that FgAreB is enriched at the target sequences that are wrapped with nucleosome, and it is able to recruit the SWI/SNF complex to the promoters of NS response genes. On contrary, the transcription repressor FgIxr1 competes with the SWI/SNF complex for binding the GATA domain of FgAreB. We further showed that NS promotes the degradation of FgIxr1 and subsequently enhances the recruitment of the SWI/SNF complex by FgAreB to the promoters of NS response genes (Fig. 7). To the best of our knowledge, this is the first report on the interplay of two TFs of a pathogenic fungus in response to environmental stress by recruiting the SWI/SNF complex.

Chromatin structure plays an important role in controlling the transcriptome and the conserved SWI/SNF chromatin-remodeling complex in eukaryotes modulates chromatin structure by mobilizing nucleosomes to activate or repress gene transcription. One unanswered question is how the remodeler decides which nucleosomes to be remodeled at a given time? Since the interaction of the complex with nucleosomal DNA has no sequence specificity[16,17], the complex should be guided to specific genomic regions by its co-operator. A growing number of

evidences have shown that various factors, including RNA polymerase II, pioneer TFs, or long non-coding RNAs (lncRNAs), might contribute to the recruitment of the SWI/SNF complex to specific genomic loci[60–62]. As an example, the yeast SWI/SNF complex has been found to be associated with a specific nucleosome by RNA polymerase II[63]. In human cancer liver stem cells, the lncTCF7 recruits the core subunits of the SWI/SNF complexes to the TCF7 promoter and promotes tumor progression[64]. Furthermore, it is well known that pioneer TFs play critical roles in opening closed chromatin regions by association with the SWI/SNF complex in various eukaryotic cells[15,65,66]. In islet β cells, the pioneer TF Pdx1 recruits the SWI/SNF complex via its Brg1 subunit to activate gene expression under high glucose conditions and interacts with its Brm subunit to repress gene transcription under low glucose condition[67]. In *Trichoderma reesei*, the XYR1 TF recruits the SWI/SNF complex to specific DNA regions by interacting with the SNF12 subunit of the complex and subsequently regulates transcription of cellulose genes[68]. More recently, we reported that the FgSR TF directly interacts with FgSwp73 and FgTaf14 of the SWI/SNF complex to enhance the recruitment of the SWI/SNF complex at the sterol biosynthesis gene promoters in *Fg* when treated with sterol biosynthesis inhibitor[66]. As more recruiters are identified, it is of immense interest to study the interplay among these factors in the future.

To regulate stress-induced gene transcription, extracellular stimuli usually activate specific recruiters, which will then associate with the SWI/SNF complex to regulate gene transcription. As an example, the yeast Rlm1 TF is phosphorylated by Slt2 (the mitogen-activated protein kinase) in the cell wall integrity pathway before recruiting the SWI/SNF complex to target genes under cell wall stress condition[17,69]. Similarly, phosphorylation of JMJD1A (a histone 3 lysine 9 demethylase) in mammalian cells by protein kinase A is likely a prerequisite for its recruitment of the SWI/SNF complex following treatment with the β-AR pan-agonist isoproterenol[65]. In parallel with these findings, we previously showed that tebuconazole (a sterol biosynthesis inhibitor) treatment activates the HOG pathway and leads to the phosphorylation of FgSR TF. Phosphorylated FgSR then recruits SWI/SNF complex

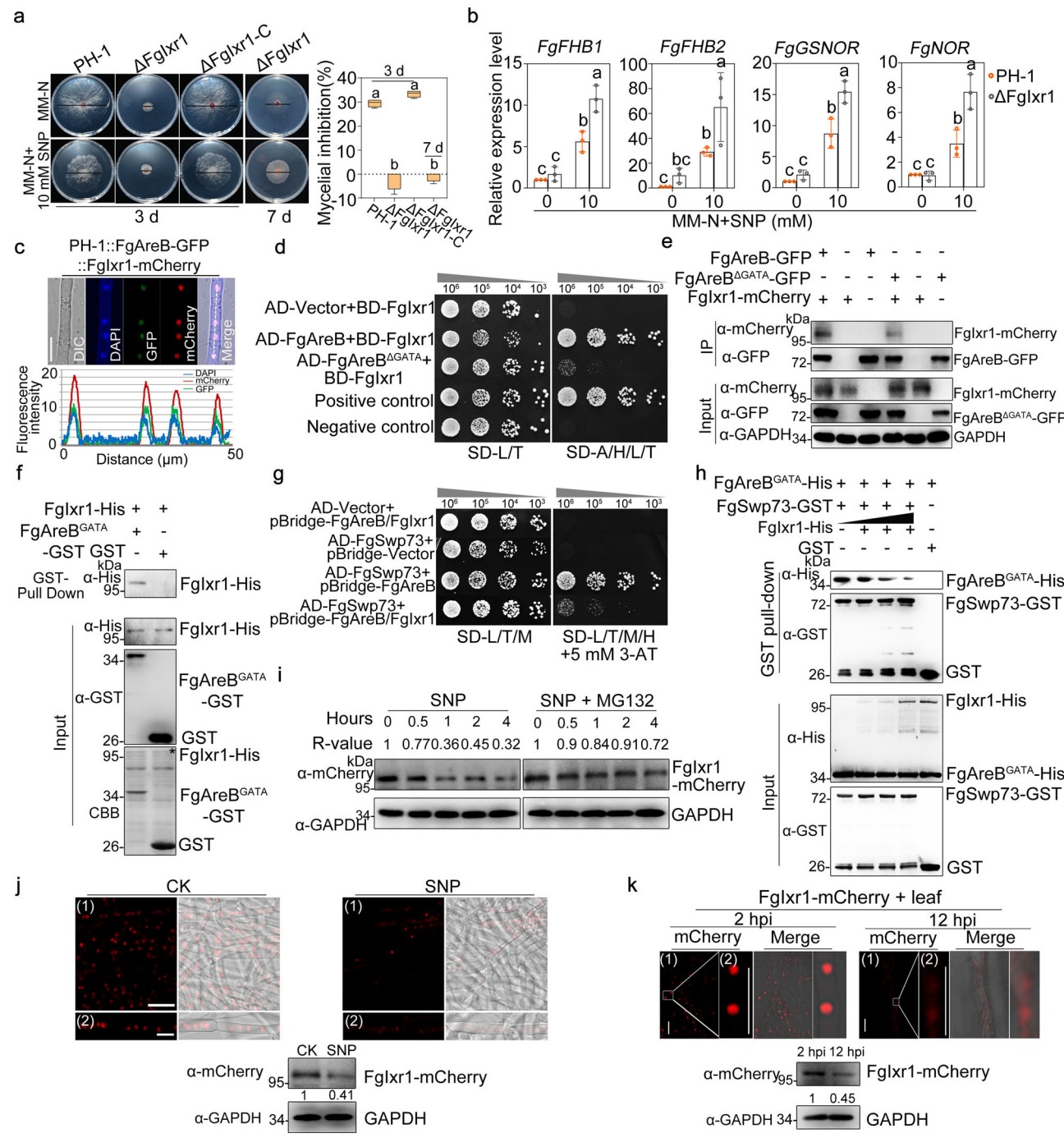

to the promoters of sterol biosynthesis genes in $Fg$[66]. In this study, we provided evidence that FgIxr1 and the FgSwp73 subunit of the SWI/SNF complex competitively interact with FgAreB. Under NS stress, degradation of FgIxr1 promotes the association of the SWI/SNF complex with FgAreB (Fig. 7). However, it remains unclear how NS leads to the degradation of FgIxr1. Since NO can induce protein S-nitrosylation, which is critical for ubiquitination-dependent protein degradation[70,71]. Our preliminary results showed that FgIxr1 was ubiquitinated and the proteasome inhibitor MG132 treatment alleviated the degradation of FgIxr1 under NS stress. It is possible that NO might promote the S-nitrosylation of FgIxr1 and thus its degradation, which requires further investigation and confirmation.

Fungi can utilize various compounds as nitrogen sources. When the preferred nitrogen sources are not available, fungi can use alternative nitrogen sources by activating genes involved in nitrogen metabolite repression (NMR). The GATA TFs have been shown to play critical roles in fungal NMR. In the budding yeast, Gln3 and Gat1 act as transcriptional activators; whereas Dal80 and Deh1 act as repressors, which suppress Gln3/Gat1-mediated transcription[72]. In filamentous fungi, only two GATA TFs, AreA, and AreB, are generally involved in nitrogen regulation. AreA primarily serves as a transcriptional activator in NMR[73]; whereas the function of AreB is much more complex. In *A. nidulans* and *Penicillium chrysogenum*, AreB negatively regulates AreA-dependent gene transcription under nitrogen starvation, indicating that AreB acts as a repressor[74,75]. In contrast, AreB not only acts as a repressor of some AreA-dependent genes but also serves as an activator for other AreA-dependent genes under nitrogen starvation in *F. fujikuroi*[73]. Similarly, AreB positively

**Fig. 5 FgIxr1 and FgSwp73 competitively interact with FgAreB. a** ΔFgIxr1 showed decreased sensitivity to 10 mM SNP. A 5-mm mycelial plug of each strain was inoculated on each MM-N plate supplemented with or without 10 mM SNP and incubated at 25 °C for three and seven days (left panel). Black lines indicate colony diameters on the plates. The percentage of mycelial inhibition was calculated (right panel). Data are shown as box plots with the interquartile range as the upper and lower confines of the box, and the median as a solid line within the box. Different letters indicate statistically significant differences according to the one-way ANOVA test ($p < 0.05$). **b** FgIxr1 activated transcription of NS response genes after SNP treatment. PH-1 and ΔFgIxr1 were cultured in YEPD for 16 h, and then subsequently incubated for 4 h in MM-N supplemented with or without 10 mM SNP. The *FgACTIN* gene used as an internal control. Data presented are the mean ± standard errors from three biological replicates ($n = 3$). Different letters represent statistically significant differences according to the one-way ANOVA test ($p < 0.05$). **c** FgIxr1 and FgAreB co-localized into nucleus. The PH-1 background strain containing FgAreB-GFP and FgSwp73-mCherry constructs was cultured in YEPD for 16 h, and then incubated in MM-N for 4 h. Fluorescence was examined under confocal microscopy. Bars: 10 μm (upper panel). Co-localization of the proteins was evaluated by line scan graph analysis (lower panel). The white dotted arrow indicates the analyzed area, and the horizontal axis indicates the distance. The experiment was repeated twice independently with similar results. **d** FgIxr1 interacted with FgAreB in the Y2H assay. Serial dilutions of the yeast cells were plated on SD-L/T/H/A. AD-vector (*pGADT7*) represents an empty vector. The experiment was repeated three times independently with similar results. **e** The GATA domain of FgAreB (FgAreB[GATA]) was required for FgAreB-FgIxr1 interaction in vivo co-immunoprecipitation (Co-IP) assay. Proteins were extracted from the strains that were cultured in YEPD for 16 h and then incubated in MM-N for 4 h. The protein samples were immunoprecipitated with anti-GFP agarose beads and immunoblotted with anti-mCherry or anti-GFP antibodies (top two panels). The lower three panels show the Input control. The experiment was repeated twice independently with similar results. **f** FgAreB[GATA] interacted with FgIxr1 in vitro in GST pull-down assay. GST or FgAreB[GATA]-GST immobilized on GST beads was incubated with FgIxr1-His proteins. The beads were washed and pelleted for immunoblotting with anti-His antibody (upper panel). The lower panel shows the Input proteins. The experiment was repeated twice independently with similar results. **g** FgIxr1 hindered with the interaction of FgSwp73 with FgAreB in the yeast three-hybrid assay. Serial dilutions of the yeast cells expressing the indicated constructs were grown on SD-L/T/H/A medium supplemented with 5 mM 3-amino-1, 2, 4-triazole (3-AT). The experiment was repeated three times independently with similar results. **h** FgIxr1 interfered with the interaction of FgAreB-FgSwp73 in the GST pull-down assay. GST or FgSwp73-GST immobilized on GST beads was incubated with FgAreB[GATA]-His proteins in presence of an increased concentration of FgIxr1-His. The beads were washed and pelleted for immunoblotting with anti-His and anti-GST antibodies. The lower panel shows the Input protein. The experiment was repeated three times independently with similar results. **i** SNP promoted the degradation of FgIxr1-mCherry. The strain ΔFgIxr1::FgIxr1-mCherry growth in YEPD for 16 h was then cultured in MM-N containing 10 mM SNP with or without 200 μM MG132 for different times. The amount of FgIxr1-mCherry was detected by western blot with anti-mCherry antibody. GAPDH serves as a loading control. The intensity of the FgIxr1-mCherry band at each time point is relative to the amount of FgIxr1-mCherry before SNP treatment (0 h). The experiment was repeated three times independently with similar results. **j** SNP treatment promoted FgIxr1-mCherry degradation. ΔFgIxr1::FgIxr1-mCherry strain was cultured in YEPD for 16 h and then incubated for 4 h in MM-N with or without 10 mM SNP. Fluorescence was examined with confocal microscopy (upper panel). Bars: 20 μm (1) or 10 μm (2). The amount of FgIxr1-mCherry was detected by western blot with anti-mCherry antibody (lower panel). GAPDH serves as a loading control. The experiment was repeated twice independently with similar results. **k** Degradation of FgIxr1-mCherry during *Fg* infection. Fluorescence of FgIxr1-mCherry was observed under confocal microscopy at 2 and 12 hpi (upper panel). Bars: 20 μm (1) or 10 μm (2). The amount of FgIxr1-mCherry during infection was detected by western blot with anti-mCherry antibody. GAPDH serves as a loading control (lower panel). The experiment was repeated twice independently with similar results.

regulates key regulatory genes mediating NMR and negatively regulates the main regulator of amino acid biosynthesis in *A. nidulans*[76]. Unexpectedly, in *N. crassa*, Asd4 (an AreB homolog) has no role in nitrogen regulation[33]. Interestingly, only AreA orthologues, but no AreB orthologue, can be identified in the basidiomycetes such as *Cryptococcus neoformans* and *Ustilago maydis*[72]. In this study, we found that FgAreB is involved in regulating NS response by recruitment of the SWI/SNF complex to the target gene promoters, although we also found that FgAreB negatively regulated transcription of numerous genes in *Fg*. It certainly will provide fascinating insights into the transcriptional mechanisms and evolution of nitrogen regulation in fungi by expanding current findings into other ascomycetes.

In nitrogen regulation, NO is an intermediate product of inorganic nitrogen assimilation and is involved in various physiological processes in both prokaryotes and eukaryotes[77]. NO acts as a transient, local, intra-, and intercellular signal molecule and plays important roles in plant-pathogen interactions, including pathogen developmental processes and virulence. An ankyrin-repeat containing protein (FgANK1) and a zinc finger TF (FgZC1) are required for pathogen-derived NO production in *Fg*. The deletion mutant of *FgANK1* was unable to produce endogenous NO during the host-sensing stage and completely lost virulence on wheat head[26]. Unlike that in FgANK1 and FgZC1 mutants, we found that NO detoxification and NS response were impaired in ΔFgAreB and ΔFgSnf5 mutants, instead of NO biosynthesis[26]. Even though FgANK1-FgZC1 and FgAreB-FgSnf5 mutants both were non-pathogenic on the wheat head, the mechanisms underneath however are discrepant. Similarly,

elimination of NO production in the rice blast fungus *M. oryzae* significantly reduced fungal pathogenesis in a compatible plant-pathogen interaction[25]. On the other hand, colonization by pathogens often results in the overproduction of NO and NO-derived molecules in host tissues. So far, most of the published data support an adverse effect of NO accumulation on fungal infection due to the potential antimicrobial effect of NO against various pathogens, including *Monilinia fructicola*, *Penicillium italicum*, *P. expansum*, *Fusarium sulphureum,* and *M. oryzae*[9,78]. On the contrary, overexpression of a NO-synthesizing enzyme in *Arabidopsis* led to the increased NO production and resulted in enhanced resistance to powdery mildew[11]. In addition, NO could trigger reprogramming of defense-related genes, production of secondary metabolites, and hypersensitive response[79]. Interestingly, the environmentally friendly antimicrobial compounds chitosan and its derivatives could trigger NO production in pathogenic fungi and strongly suppress fungal virulence[80,81]. Therefore, from the practical application point of view, manipulation of NO production may represent a promising strategy for the management of plant disease in sustainable agriculture.

## Methods
**Strains and fungal growth determination**. The wild-type strain PH-1 (NRRL 31084) of *Fg* was used as a parental strain. Mycelial growth of the PH-1 and its derivative strains were assayed on MM-N with or without SNP (S118460, Aladdin, Shanghai, China). Three replicates were included for each strain, and experiments were repeated three times.

**Mutant generation**. Gene deletion mutants of *Fg* were constructed using poly-ethylene glycol (PEG) mediated protoplast transformation method[82]. Briefly, to

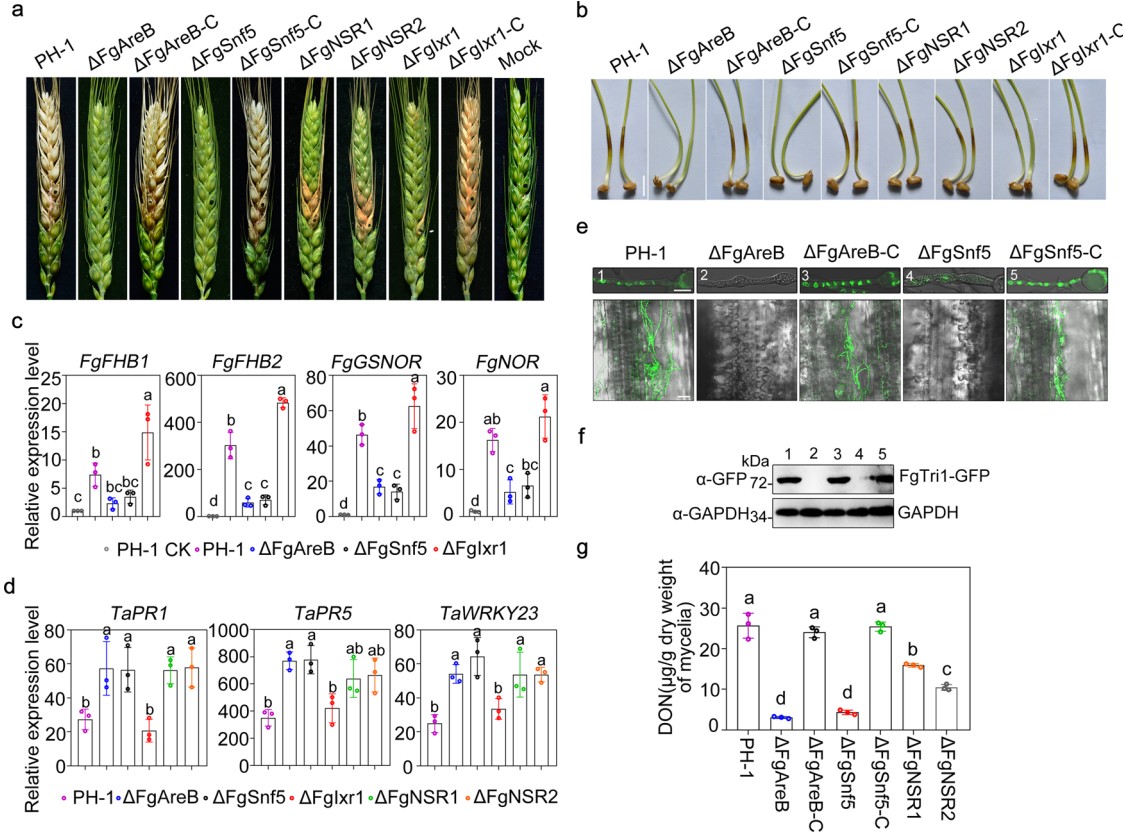

**Fig. 6 FgAreB, FgSnf5, and FgNSRs contribute to the virulence of *F. graminearum*. a** Virulence of PH-1, ΔFgAreB, ΔFgAreB-C, ΔFgSnf5, ΔFgSnf5-C, ΔFgNSR1, ΔFgNSR2, ΔFglxr1, and ΔFglxr1-C strains on wheat head. The infected wheat heads were photographed 15 day-post-inoculation (dpi). The black dot represents the inoculated site on each wheat head. Sterile water used as control (Mock). The experiment was repeated twice independently with similar results. **b** Virulence of PH-1, ΔFgAreB, ΔFgAreB-C, ΔFgSnf5, ΔFgSnf5-C, ΔFgNSR1, ΔFgNSR2, ΔFglxr1, and ΔFglxr1-C strains on wheat coleoptiles. Representative images were taken three days after inoculation. Bars: 1 cm. The experiment was repeated three times independently with similar results. **c** Expression of NS response genes in ΔFgAreB and ΔFgSnf5 was repressed after infection of wheat seedling leaves at 12 hpi. The expression level of each NS response gene in the PH-1 incubated in MM-N (PH-1 CK) was set to 1. Data presented are the mean ± standard errors from three biological replicates (*n* = 3). Different letters indicate statistically significant differences according to the one-way ANOVA test (*p* < 0.05). **d** Expression of wheat *TaPR1*, *TaPR5*, and *TaWRKY23* genes was evaluated after inoculation with ΔFgAreB, ΔFgSnf5, ΔFgNSR1, or ΔFgNSR2. The transcription of *TaPR1*, *TaPR5,* and *TaWRKY23* in wheat seedling leaves was determined by real-time quantitative PCR at 12 h post-inoculation. *TaTUBULIN* gene used as an internal control. Data presented are the mean ± standard errors from three biological replicates (*n* = 3). Different letters represent statistically significant differences according to the one-way ANOVA test (*p* < 0.05). **e** Toxisome formation was restrained in ΔFgAreB and ΔFgSnf5. ΔFgAreB and ΔFgSnf5 showed the defect in toxisome formation under DON induction medium (TBI) (upper panel). Toxisome formation was observed using the FgTri1-GFP marker under confocal microscopy. Bars: 10 μm. Defect of toxisome formation in ΔFgAreB and ΔFgSnf5 after infection on wounded seedling leaves at 48 h (lower panel). Bars: 50 μm. The experiment was repeated twice independently with similar results. **f** FgTri1-GFP protein level was undetectable in ΔFgAreB and ΔFgSnf5 cultured in TBI. The intensity of FgTri1-GFP was determined by immunoblot assay using an anti-GFP antibody. FgGAPDH used as the loading control. The experiment was repeated twice independently with similar results. **g** The mutants ΔFgAreB, ΔFgSnf5, ΔFgNSR1 or ΔFgNSR2 showed reduced deoxynivalenol (DON) production in TBI. Each strain was determined for DON production after growth in TBI for three days. Data presented are the mean ± standard errors from three biological replicates (*n* = 3). Different letters represent statistically significant differences according to the one-way ANOVA test (*p* < 0.05).

obtain *Fg* protoplasts, fresh mycelia were treated with driselase (D9515, Sigma, MO, USA), lysozyme (RM1027, RYON, Shanghai, China), and cellulose (RM1030, RYON, Shanghai, China). Primers used to amplify the flanking sequences for each gene are listed in Supplementary Data 11. Putative gene deletion mutants were confirmed by PCR assays, and the FgAreB deletion mutant was further confirmed by a Southern blot assay.

To construct mutant complementation strain, the FgAreB-GFP fusion fragment was co-transformed with XhoI-digested pYF11 into the yeast strain XK1-25 using the Alkali-Cation™ Yeast Transformation Kit (MP Biomedicals, Solon, USA) to generate a FgAreB-GFP fusion vector. FgSnf5-GFP fusion cassette was similarly constructed. Complementation of ΔFgAreB and ΔFgSnf5 with FgAreB-GFP and FgSnf5-GFP vector, respectively, was achieved using geneticin (*G418*) as the second selectable marker.

**Pathogenicity, toxisome induction, and DON production assays**. To assess pathogenicity on flowering wheat heads, conidia harvested from four-day-old cultures of *Fg* strains were resuspended in 0.01% (v/v) Tween 20 and adjusted to

$10^5$ conidia/ml. Ten microliters aliquot of a conidial suspension was injected into a floret in the middle spikelet of a wheat head of the susceptible cultivar Jimai22 at early anthesis. There were 20 replicates for each strain. After fifteen days post-inoculation, infected spikelets in each inoculated wheat head were recorded.

Three-day-old seedlings of susceptible wheat (*Triticum aestivum*) cultivar Jimai22 were used for coleoptile infection. Three days after sowing, the top 2-mm to 3-mm of the coleoptiles was removed and the wounding top was directly inoculated with a 5-mm-diameter fresh mycelial plug. After inoculation, the seedlings were grown in a growth chamber at 25 °C and 95% humidity for three days before examination for the infection.

To observe toxisome formation, strains labeled with Tri1-GFP were constructed as previously described[55]. Each Tri1-GFP labeled strain was cultured in TBI. After incubation for two days at 28 °C and 150 rpm in the dark, toxisome formation was indicated by the FgTri1-GFP marker and observed under a Zeiss LSM780 confocal microscope (Gottingen, Niedersachsen, Germany). To quantify DON production, each strain was grown in TBI medium at 28 °C for seven days in a shaker (150 rpm) in the dark. DON Quantification Kit Wis008 (Wise Science, Zhenjiang, China) was used to quantify the DON production for each sample.

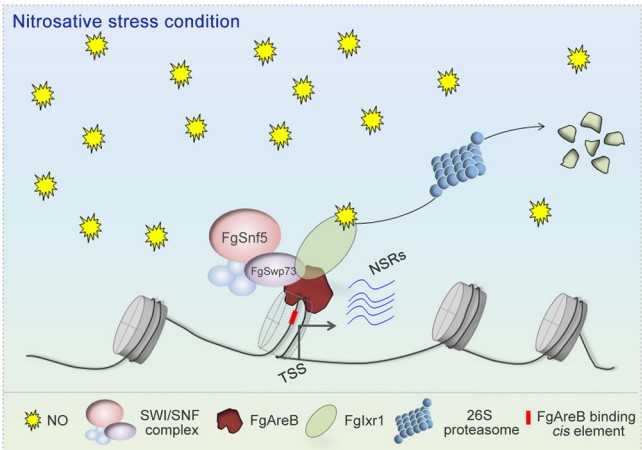

**Fig. 7 A proposed model for transcriptional regulation mechanism in response to nitrosative stress in *F. graminearum*.** Nitrosative stress leads to degradation of transcription repressor FgIxr1 by the 26S proteasome. The degradation of FgIxr1 promotes the recruitment of the SWI/SNF complex by the pioneer transcription factor FgAreB, subsequently activates the expression of NS response genes and eventually accelerates NO metabolism. NO (<u>N</u>itric <u>O</u>xide), NSRs (<u>N</u>itrosative <u>S</u>tress <u>R</u>esponse genes); TSS (<u>T</u>ranscription <u>S</u>tart <u>S</u>ite).

**Chromatin immunoprecipitation (ChIP)-qPCR analyses.** ChIP was performed based on the published protocol with modifications[83]. Briefly, fresh mycelia of each sample were cross-linked with 1% formaldehyde for 10 min, and then the reaction was stopped with 125 mM glycine for 5 min. Subsequently, samples were grounded in liquid nitrogen and suspended in lysis buffer with protease inhibitor (Sangon Co., Shanghai, China). DNA was sheared into 200–500 bp fragments with 30 s on and 30 s off in a Bioruptor Plus (Diagenode, UCD-300). After centrifugation, the supernatant was diluted with 10×ChIP dilution buffer (1.1% Triton X-100, 1.2 mM EDTA, 16.7 mM Tris-HCl, pH 8.0 and 167 mM NaCl). Immunoprecipitation was conducted using the Rabit polyclonal anti-GFP (ab290, Abcam, Cambridge, UK; 1:500 dilution) antibody or Rabit monoclonal anti-H3 (ab1791, Abcam, Cambridge, UK; 1:500 dilution) antibody together with the protein A agarose beads (sc-2001, Santa Cruz, CA, USA). It should be noted that the specificity of ChIP grade GFP antibody to FgAreB-GFP, FgSnf5-GFP, and H3 antibody used for ChIP-qPCR assays in this study was verified by western blot (Supplementary Fig. 13). The beads were subsequently washed by low salt wash buffer, high salt wash buffer, LiCl wash buffer, and TE buffer[66]. The immunoprecipitated complexes were then eluted from beads with freshly prepared elution buffer (1% SDS, 0.1 M NaHCO₃). The resulting complexes were treated with 2 μl RNase/sample at 37 °C for 2 h, reversed cross-linking, and digested with proteinase K at 45 °C for 2 h. For each sample, an equal volume of phenol/chloroform/isoamyl alcohol was added to precipitate DNA at −20 °C for 3 h or overnight. The ChIP-enriched DNA was used for quantitative PCR analysis using ChamQ SYBR qPCR Master Mix (Vazyme, Q311-02, Nanjing, China) along with the corresponding primers (Supplementary Data 11). Relative enrichment of each gene was determined by quantitative PCR and calculated by normalizing the value of the immunoprecipitated sample with that of the input. The experiment was repeated three times.

**Yeast two-hybrid (Y2H) and Yeast three-hybrid (Y3H) assays.** To construct plasmids for Y2H analysis, the coding sequence of each gene was amplified from cDNA of the PH-1 with corresponding primer pairs (Supplementary Data 11). The cDNA fragments were cloned into the yeast GAL4-binding domain vector pGBKT7 and GAL4-activation domain vector pGADT7 (Clontech, Mountain View, CA, USA), respectively. Pairs of Y2H plasmids were co-transformed into *S. cerevisiae* strain Y2H Gold following the lithium acetate/single-stranded DNA/polyethylene glycol transformation protocol. The plasmid pair pGBKT7-53 and pGADT7-T used as a positive control. The plasmid pair pGBKT7-Lam and pGADT7-T used as a negative control. Transformants were grown at 30 °C for three days on synthetic medium (SD) lacking Leu and Trp, and then serial dilutions of yeast cells (cells/ml) were transferred to SD without His, Leu, Trp, and Ade to assess protein-protein interaction. Three independent experiments were performed to confirm each Y2H result.

To conduct Y3H assay, FgAreB fused with GAL4 DNA-binding domain was cloned into MCS I of pBridge, and FgIxr1 was cloned into MCS II of pBridge. Subsequently, pBridge-FgAreB-FgIxr1 and AD-FgSwp73 vectors were co-transformed into the Y2H gold yeast strain. The resulting strain was cultured on a medium lacking Leu, Trp, His, and Met and supplemented with 5 mM 3-amino-1, 2, 4-triazole (3-AT)[84,85].

To identify FgAreB-interacting proteins, FgAreB was cloned into the yeast vector pGBKT7 (Clontech, Mountain View, CA, USA). An *Fg* cDNA library was constructed in the Y2H vector pGADT7 using total RNAs extracted from fresh mycelia. The Y2H Gold co-transformed with cDNA library and FgAreB-pGBKT7 was directly selected on SD-Trp-Leu-His. Yeast transformants containing cDNA clones interacting with FgAreB were sequenced (Supplementary Data 2).

**Co-IP assays.** Constructs of targeted genes fused with different markers (GFP, 3×FLAG, or mCherry) were verified by DNA sequencing and transformed in pairs into PH-1. Transformants expressing a pair of fusion constructs were confirmed by western blot analysis. For Co-IP assays, fresh mycelia (500 mg) of each strain were finely grounded and suspended in 1 ml of extraction buffer containing 10 μl of protease inhibitor cocktail (Sangon Co., Shanghai, China). After homogenization with a vortex shaker, the lysates were centrifuged at 10,000×g for 20 min at 4 °C and the supernatants were incubated with anti-GFP (ChromoTek, Martinsried, Germany) agarose. Proteins eluted from agarose were analyzed by western blot with Mouse monoclonal anti-FLAG (A9044, Sigma, St. Louis, MO, USA, 1:5000 dilution), Rabit polyclonal anti-GFP (ab32146, Abcam, Cambridge, UK, 1:5000 dilution), and Mouse monoclonal anti-mCherry (ab125096, Abcam, Cambridge, UK, 1:2000 dilution) antibodies. Samples were also detected with Mouse monoclonal anti-GAPDH antibody (EM1101, Hua An Biotech. Ltd., Hangzhou, China, 1:5000 dilution) as a reference. After inoculation with secondary antibody Goat polyclonal anti-rabbit IgG-HRP (HA1001, Hua An Biotech. Ltd., Hangzhou, China, 1:5000 dilution) or Goat polyclonal anti-mouse IgG-HRP (HA1006, Hua An Biotech. Ltd., Hangzhou, China, 1:5000 dilution), respectively, chemiluminescence was detected. All blots were imaged by Image Quant LAS4000 mini (GE Healthcare, Chicago, USA).

**GST pull-down assay.** To conduct the GST pull-down assay, cDNA containing full-length FgSwp73 and the FgAreB^GATA domain (encoding the N terminal 1-165 amino acids of FgAreB) were amplified by PCR and cloned into pGEX-4T-3 and pGEX-6p-1, respectively, to generate constructs for GST-tagged proteins. In addition, cDNAs encoding FgIxr1 and FgAreB^GATA were amplified and cloned into pET32a to generate constructs for His-tagged proteins. Each GST-tagged or His-tagged protein expressed in the *Escherichia coli* strain BL21 was purified.

Co-precipitation of FgAreB^GATA-His with FgSwp73-GST or FgAreB^GATA-GST with FgIxr1-His was examined by western blotting before (input) and after affinity purification (pull-down) using GST Agarose (Thermo Fisher Scientific, Waltham, USA). The GST protein used as a negative control. Interactions among FgSwp73-GST, FgAreB^GATA-GST, FgAreB^GATA-His, and FgIxr1-His proteins were detected by using Mouse monoclonal anti-GST (M0807, Hangzhou Hua An Biotechnology Co., Ltd., 1:1000 dilution) and Mouse monoclonal anti-His (ab18184, Abcam, Cambridge, MA, USA, 1:1000 dilution) antibodies.

To assay, the competition between FgSwp73 and FgIxr1 for binding FgAreB^GATA, different concentrations of FgIxr1-His were mixed with FgAreB^GATA-His, and the mixtures were co-incubated with FgSwp73-GST at 4 °C for 2 h. The protein complexes were then pull-down with the GST beads by washing six times with 1×TBS to avoid nonspecific binding and then eluted by 5×SDS buffer. The resulting protein samples were boiled for 10 min and analyzed by western blotting using anti-His and anti-GST antibodies. All blots were imaged by Image Quant LAS4000 mini (GE Healthcare, Chicago, USA). The experiment was performed three times.

**Electrophoretic mobility shift assay (EMSA).** To determine the binding ability of FgAreB with its *cis*-element, FgAreB^GATA was amplified and cloned into pET32a vector to generate the FgAreB^GATA-His fusion construct. The resulting construct was transformed into *E. coli* strain BL21. The recombinant FgAreB^GATA-His protein expressed in BL21 was purified by Ni sepharose beads and eluted by imidazole. Probe DNAs labeled with biotin at the 3´ end and their reverse complementary chains were synthesized by ThermoFisher Scientific Shanghai Agency. EMSAs were performed with LightShift™ Chemiluminescent EMSA Kit (Thermofisher Company, USA) according to the manufacturer's instruction[86]. Briefly, purified FgAreB^GATA-His and biotin-labeled or competitive probes were incubated in a 20 μl reaction mixture for 20 min at room temperature. The reactions were electrophoresed on 6% polyacrylamide gel in 0.5×TBE 100 V on ice for about 90 min and transferred to a positively charged nylon membrane (Millipore, USA). Signals were detected by Image Quant LAS4000 mini (GE Healthcare, Chicago, USA). The experiment was performed three times.

**MNase assay.** MNase assay was conducted using a previously reported protocol[87]. Briefly, to isolate fungal nuclei, fresh mycelia of each strain were frozen in liquid nitrogen and grounded to a fine powder using a mortar and pestle. The resulting powdered mycelia (0.6 g) were mixed with 1 ml of lysis buffer (250 mM sucrose, 25% (v/v) glycerol, 2 mM MgCl₂, 20 mM KCl, 20 mM Tris-HCl (pH 7.5), 5 mM DTT) and incubated on ice for 5 min with constant stirring. The homogenates were then filtered through cheesecloth and subsequently centrifuged at 1500 × g for 15 min at 4 °C. The pellets were re-suspended in 1 ml NEB1 buffer (Nuclei Extraction Buffer) (20 mM Tris-HCl (pH 7.5), 0.2% (w/v) TritonX-100, 25% (v/v) glycerol, 2.5 mM MgCl₂) and centrifuged at 15,000×g for 10 min at 4 °C. This step was

repeated five times. The resulting pellets were re-suspended in 1 ml NEB2 buffer (20 mM Tris-HCl (pH 7.5), 0.5% (w/v) TritonX-100, 250 mM Sucrose, 10 mM MgCl$_2$, 5 mM β-mercaptoethanol), layered onto 1 ml of NEB3 buffer (20 mM Tris-HCl (pH 7.5), 0.5% (w/v) TritonX-100, 1.7 M Sucrose, 10 mM MgCl$_2$, 5 mM β-mercaptoethanol), and centrifuged at $16,000 \times g$ for 45 min at 4 °C. The final nuclei pellet was suspended in MNase reaction buffer.

To conduct MNase digestion, an equal portion of nuclei isolation (160 μl) of each sample was mixed with 320 μl with MNase reaction buffer and treated with 4 μl MNase enzyme (Takara, Beijing, China). The same amount of nuclei isolation without MNase treatment used as an undigested control. Samples were incubated at 37 °C for 8 min, and the reaction was terminated by adding 50 μl of stop buffer (50 μl 10% SDS, and 40 μg proteinase K) at 60 °C for 1 h. Each resulting sample was then treated with 1 U RNase (10 μg/μl) at 37 °C for 1 h and stored at 4 °C overnight. DNAs from each sample were extracted by using the phenol-chloroform-isoamyl alcohol method and re-suspended in 50 μl water. The resulting DNA samples were used for quantitative PCR with multiple pairs of primers spanning the tested region (Supplementary Data 11). The resulting amplicons having an average size of 100 bp with 20 bp overlap were used for analyzing nucleosome occupancy[87].

**Determination of NS**. Nitrotyrosine was determined as previously described[28]. Briefly, 100 mg of mycelia were suspended in 1 ml alkaline lysis buffer (25 mM Tris, 100 mM SDS and 128 mM NaOH) and were homogenized using Tissuelyser-48 (Shanghai Jingxin Co., Shanghai, China). The extracted proteins from each sample were separated on a polyacrylamide gel and transferred to a polyvinylidene fluoride (PVDF) membrane using a Transblot TurboTransfer System (Bio-Rad Laboratories, Shanghai, China). PVDF membrane was blocked with 5% nonfat dry milk in 0.1% Tween-20 (TBST) buffer (0.1% Tween-20,150 mM NaCl, 100 mM Tris, pH 7.5). The membrane was then probed with Rabbit polyclonal anti-nitrotyrosine antibody at 1:1000 dilution (A-21285, Thermo Scientific) and washed five times with TBST. The membrane was further incubated with horseradish peroxidase (HRP)-conjugated goat anti-rabbit IgG secondary antibody (1:2000 dilution) and washed five times with TBST. Immuno-reactive proteins were visualized using Image Quant LAS4000 mini (GE Healthcare, Chicago, USA). In addition, each protein sample separated on SDS-PAGE gels was stained with coomassie-blue, which serves as a loading control.

Malondialdehyde was determined by using the TBARS assay kit (Cayman, Hamburg, Germany). Briefly, fresh mycelia (200 mg) of each sample were homogenized in HN buffer (20 mM NaCl, 50 mM HEPES, pH 7.5). Samples were then centrifuged at $15,000 \times g$ for 1 min to pellet cell debris, and the supernatant (100 μl) was mixed with 100 μl 10% SDS. Subsequently, 4 ml of MDA colorimetric reagent was added to the mixture, boiled for 1 h, and quenched on ice for 10 min. The final solutions were read at 450 nm by Varioskan Flash (Thermo Scientific). The concentration of malondialdehyde was determined using the standard included in the TBARS assay kit according to the manufacturer's instruction. Each experiment was repeated three times.

**Staining and microscopic examination**. Cellular ROS accumulation in fungal infected plant cells was analyzed by using DAB (Sigma, D-8001) staining method[88]. Briefly, infected tissues were immersed in DAB solution (1 mg/ml, pH 3.8) at room temperature for 8 h and destained with decolorant solution (ethanol: acetic acid = 94: 4, v/v) for 1 h. The DAB-stained plant tissues were examined under a Zeiss LSM780 confocal microscope (Gottingen, Niedersachsen, Germany).

To assess NO production, fungal hyphae or plant tissue were stained using 10 μM DAF-FM DA (Sigma, Shanghai, China). To confirm genuine reaction between DAF-FM DA and NO, a cell-permeant NO scavenger cPTIO (Sigma) at 100 μM was included as a control. DAF-FM DA combined with or without cPTIO was directly applied towards plant tissues with or without *Fg* infection. Plant tissues or fungal hyphae were stained with DAF-FM DA dye for 30 min at room temperature and visualized in a bight/fluorescence field of view under a Zeiss LSM780 confocal microscope (Gottingen) at the excitation/emission wavelengths of 488/525 nm. Each experiment was repeated three times.

**ATAC-seq analysis**. ATAC-seq analysis was performed as described previously by Buenrostro et al[89]. Briefly, fungal nuclei extracted from 0.1 g fresh *Fg* mycelia of each strain were used for transposition reaction by 2.5 μl of Nextera Tn5 transposase (Illumina, California, United States) at 37 °C for 30 min. After the transposition reaction, DNAs were purified using a Qiagen MinElute Purification Kit (Qiagen, Hilden, Germany). Each purified DNA sample was amplified using Nebnext High-Fidelity 2×PCR Master Mix (New England Biolabs, Massachusetts, United States) with corresponding primers (Supplementary Data 11) for 15 cycles for constructing an amplified library. The resulting amplified libraries were subjected to purification using Qiagen MinElute Purification Kit and sequenced by Wuhan Igenebook Biotechnology Co., Ltd., China.

To analyze ATAC-seq data, adapters and low-quality reads were filtered out through Trimmomatic (version 0.38). Clean reads were mapped to the *Fg* genome by Hisat2 (version 2.1.0), allowing up to two pair base mismatches for each read. The Samtools (version 1.3.1) were used to remove potential PCR duplicates, and MACS2 software (version 2.1.1.20160309) was used to call peaks using the

following parameters: no model; shift-100; ext size 200; model fold, 5, 50; *q* value, 0.05. The diffpeak was identified using Diffbind (version 1.16.3). If the summit of a peak located within 2000 bp around TSS (Transcription Start Site) of a gene, the peak will be assigned to that gene. Q values obtained by Fisher's exact test were adjusted with FDR (False Discovery Rate) for multiple comparisons. Heatmap of regulatory region distribution was obtained using Deeptool (version 3.2.1). The experiment was repeated twice for each test strain.

**Reporting summary**. Further information on research design is available in the Nature Research Reporting Summary linked to this article.

## Data availability
Relevant data supporting the findings of this study are available in this article and its Supplementary Information files. The ATAC-Seq and RNA-Seq data have been deposited in the NCBI BioProject database with accession code PRJNA648257. Source data are provided with this paper.

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

## Acknowledgements

This research was supported by the Key Project of National Natural Science Foundation of China (31930088), China Agriculture Research System (CARS-3-1-29), and National Natural Science Foundation (31801675).

## Author contributions

Y.J., Z.L. designed the experiments; Y.J., Z.L., and H.W. conducted the experiments; Z.M. directed the project; Y.J., Z.L., Y.C., Y.Y., Y.Z., and Z.M. analyzed data and wrote the manuscript. All authors read and approved the manuscript.

## Competing interests

The authors declare no competing interests.
