## [Peer Review File · Nature Communications]

REVIEWER COMMENTS

Reviewer #1 (Remarks to the Author):

The manuscript by Jian et al describes in very intricate detail key molecular players in the response of the fungus *Fusarium graminearum* to exogenous nitric oxide. This study is very elegant and thorough and reveals some very new findings about the important processes of plant pathogen virulence most likely with broad applicability to other fungal pathogens of plants (and possibly animals).

I have almost no criticisms of the work and think it should be accepted in its current form.

minor query: line 346 RROSOS: do you mean ROS?

Reviewer #2 (Remarks to the Author):

Manuscript 270432_0 "Interplay of two transcription factors for recruitment of chromatin remodeling complex modulates fungal nitrosative stress response" by Jian et al.

The manuscript by Jian et al. describes the analysis of the transcription factors FgAreB and FgIxr1 from the filamentous ascomycete *Fusarium graminearum*, and their role in recruiting the SWI/SNF chromatin remodeling complex to promoters of genes involved in the fungal reaction to nitrosative stress.

The study is extremely interesting and very comprehensive in that it uses multiple techniques to show that the two transcription factors have opposing roles in the stress regulation process, the different interactions of the transcription factors and of FgAreB and the Swp73 subunit of the SWI/SNF complex, the differential binding of AreB to promoters of target genes, and the consequences of deletions of the transcription factors and several subunits of the SWI/SNF complex on gene expression and the fungal phenotype. The results indicate that FgAreB acts as pioneer transcription factor that recruits the SWI/SNF complex to specific target genes under nitrosative stress conditions.

Overall, the study is highly interesting to researchers working on the regulation of chromatin structure and its effects on gene expression as well as for researchers working on pathogenicity mechanisms of fungi, especially since not much is known yet on the role of chromatin regulation in fungal pathogenicity.

There are just some points where the manuscript could be improved:

1. Several Figures show fluorescence microscopy pictures that are extremely difficult to see, both in the pdf as well as in a printout, e.g. Fig. 1B, 1F, 2C, 3G, 5J, S1A. Visibility should be improved, e.g. by converting to gray scale and/or inverting the color scale of the picture.
2. Fig. S2: The accession numbers of the proteins used to generate the phylogenetic tree should be given, otherwise it is difficult to know which of the usually several GATA factors in each species was used to construct the tree.
3. Fig. S3B and S10A: It should be described in more detail what was amplified in the PCRs.
4. Fig. 2H and text lines 195-197: It should be described in more detail under which conditions the samples for the MNase assay were grown, e.g. were these conditions for nitrosative stress or not? Also, it might be better to rephrase the text describing this figure, because the presence of a nucleosome at the putative FgAreB binding site does not in itself implicate that FgAreB is a pioneer factor. It might be possible that other factors are required to remove the nucleosome, and the fact that FgAreB is required for recruitment of the SWI/SNF complex to promoters and not the other way round, which is a better indicator for FgAreB acting as a pioneer factor, is shown only later.

5. Fig. S5G: It is not clear what is shown in the left vs. the right panel.
6. Fig. S6B: Two strains are labelled as FgSwp82, and none as FgSnf5, there might be a labelling problem.
7. Text lines 285-288 and Fig. S9E: The peaks for the NOR gene regions (FGSG_10200 and FGSG_11585) do not look smaller or otherwise different from the wild type, contrary to what is described in the text.
8. Fig. S9C: How was correlation calculated?
9. Fig. S9D: The letters are much too small to be read.
10. Fig. 4F: What about the opposite, i.e. peaks that are more accessible in the two mutants compared to the wild type? Are there any, and if so, is there overlap between the mutants?
11. Text line 294 and Fig. 5A: It is stated that the FgIxr1 mutant shows decreased sensitivity to SNP, but in the pictures only a comparison to wild type is shown, compared to which the mutant has increased sensitivity.
12. The text and Figures describe yeast-three-hybrid assays, but these are not described in the Materials and Methods sections.
13. Lines 364 ff.: Please describe in more detail the function of the FgTri1 protein and the text for toxisomes that was performed.
14. Line 387: The statement that this study demonstrates that FgAreB binds nucleosomal DNA might be too strong. The actual binding assays (EMSAs) were done in vitro with non-nucleosomal DNA. For the ChIP assays, it is not quite clear from the descriptions of the experiments if the status of the DNA (which would be depending on growth conditions etc.) corresponds to the one for the experiments shown e.g. in Fig. 2H.
13. The RNA-seq and ATAC-seq data generated in this study should be submitted to a public database (e.g. NCBI GEO or SRA) and accession numbers need to be included in the manuscript.

Reviewer #3 (Remarks to the Author):

Referee opinion on "Interplay of two transcription factors for recruitment of the chromatin remodeling complex modulates fungal nitrosative stress response" by Jian et al.

The work is potentially of high interest but most of the conclusions drawn by the authors are not based on experimental evidence or on experiments lacking appropriate controls. Generally, a large amount of data does not replace the requirement for high quality of data.

Specific comments:

1. All experiments are performed under nitrogen starvation conditions only and no data are available under normal metabolic conditions – the explanation somewhere in the text is that "nitrogen sources lead to NO accumulation". In this case all the effects should be even more pronounced. Clearly, the influence of a number of different nitrogen sources on NO generation and the function of the TFs and SWI/SNF needs to be tested (wild type and mutants responses).
2. The authors screened 1500 deletions mutants? No description of the screen or the mutant library is provided
3. Authors state that it is FgAreB that binds to NS promoters – controls of mutants in other GATA-factor genes (AreA, SreA, etc.) are missing. No conclusions can be drawn from the present EMSAs (from line 176 onwards).
4. It is clear that AreB seems to be involved – but if it is the "pioneer TF" initializing PIC assembly

and SWI/SNF recruitment remains enigmatic from the experimental data. In spite of this, the authors develop a wild hypothesis on NS-gene regulation by AreB and competition by Ixr1 that is totally unjustified. No direct in vivo competition experiments (e.g. sequential ChIP-on-ChIP) showing competition are provided.

5. Line 64-Intro: it is not true that TF have an initializing role in binding to closed chromatin – in contrary, they usually bind to NFRs and then recruit what it takes to open/remodel chromatin.

6. Lines 100-102: solvent controls are missing.

7. Line 114: growth test under these conditions not valid – nitrogen supply also needed and biomass measurements; colony extension does not represent biomass in fungi grown on solid media.

8. Line 137: in many case the GATA domain overlaps with the NLS – is this the case also here? A putative NLS search result should be included

9. Line 167: statement needs triple mutant addition to experimental design

10. Fig 2 (lines ~190) : controls of other GATA factors missing, control without SNP missing, - statement invalid, all the effects could be indirect

11. Co-IP and Y2H data contradictory for Snf5 and Snf2

12. Line 255: areB presence is not SNP dependent – this is a contradiction to statements above

13. Line 280: MNase assays as done here are not quantitative – statement needs to be removed or experimental design needs to be changed

14. Line 267: conditions for RNA seq are not described

15. Line 298: Ixr1 tagging is not described

16. Line 304: conditions are not described for nuclear positioning observations.

17. Line 308: statement invalid: it could be other GATA factors that interact with their GATA domains – shown that this domain is crucial for interaction – so it could easily be other factors that pull this TF into the nucleus

18. Fig 5/lane 330: how significant are these statements based fluorescence measurements? What is the statistics, which hyphae are considered?

19. Discussion lines 485ff: I wonder why in all these interaction screens the known TFs ANK1 and ZC1 are not appearing? This should be discussed.

20. Methods: how were the ChIP antibodies validated? Solvent control for chemo-biological experiments generally missing

Point-by-point responses:

Reviewer #1

The manuscript by Jian et al describes in very intricate detail key molecular players in the response of the fungus *Fusarium graminearum* to exogenous nitric oxide. This study is very elegant and thorough and reveals some very new findings about the important processes of plant pathogen virulence most likely with broad applicability to other fungal pathogens of plants (and possibly animals).

I have almost no criticisms of the work and think it should be accepted in its current form.

minor query: line 346 RROSOS: do you mean ROS?

Response: Thanks for pointing out this. Corrected.

Reviewer #2 (Remarks to the Author):

Manuscript 270432_0 "Interplay of two transcription factors for recruitment of chromatin remodeling complex modulates fungal nitrosative stress response" by Jian et al.

The manuscript by Jian et al. describes the analysis of the transcription factors FgAreB and FgIxr1 from the filamentous ascomycete *Fusarium graminearum*, and their role in recruiting the SWI/SNF chromatin remodeling complex to promoters of genes involved in the fungal reaction to nitrosative stress.

The study is extremely interesting and very comprehensive in that it uses multiple techniques to show that the two transcription factors have opposing roles in the stress regulation process, the different interactions of the transcription factors and of FgAreB and the Swp73 subunit of the SWI/SNF complex, the differential binding of AreB to promoters of target genes, and the consequences of deletions of the transcription factors and several subunits of the SWI/SNF complex on gene expression and the fungal phenotype. The results indicate that FgAreB acts as pioneer transcription factor that recruits the SWI/SNF complex to specific target genes under nitrosative stress conditions.

Overall, the study is highly interesting to researchers working on the regulation of chromatin structure and its effects on gene expression as well as for researchers working on pathogenicity mechanisms of fungi, especially since not much is known yet on the role of chromatin regulation in fungal pathogenicity.

There are just some points where the manuscript could be improved:

1. Several Figures show fluorescence microscopy pictures that are extremely difficult to see, both in the pdf as well as in a printout, e.g. Fig. 1B, 1F, 2C, 3G, 5J, S1A. Visibility should be improved,

e.g. by converting to gray scale and/or inverting the color scale of the picture.

Response: Sorry about that and thanks for your suggestions. We have improved the visibility of the fluorescence microscopy pictures as suggested. See Figs. 1b, 1f, 2c, 3g, 5j, and S1a.

2. Fig. S2: The accession numbers of the proteins used to generate the phylogenetic tree should be given, otherwise it is difficult to know which of the usually several GATA factors in each species was used to construct the tree.

Response: The accession numbers of the proteins were added as suggested. See Fig. S2b.

3. Fig. S3B and S10A: It should be described in more detail what was amplified in the PCRs.

Response: Details were added in the Figure and Figure legends for Fig. S3b, S4a, and S10a, as well as primer pairs used (Table S12). See Line 1371-1374, Line 1402-1407, and Line 1507-1510.

4. Fig. 2H and text lines 195-197: It should be described in more detail under which conditions the samples for the MNase assay were grown, e.g. were these conditions for nitrosative stress or not? Also, it might be better to rephrase the text describing this figure, because the presence of a nucleosome at the putative FgAreB binding site does not in itself implicate that FgAreB is a pioneer factor. It might be possible that other factors are required to remove the nucleosome, and the fact that FgAreB is required for recruitment of the SWI/SNF complex to promoters and not the other way round, which is a better indicator for FgAreB acting as a pioneer factor, is shown only later.

Response: Yes, you are correct. The presence of nucleosomes at the putative FgAreB binding site does not in itself implicate that FgAreB is a pioneer factor. We have rephrased the text. See line 205-214. Detailed conditions were also provided in the Figure legend. See Line 1125 and Line 1129-1131.

5. Fig. S5G: It is not clear what is shown in the left vs. the right panel.

Response: Sorry for the confusion. The left and right panels show the association of FgAreB with the two key SWI/SNF complex subunits FgSnf5 and FgSnf2, respectively. To be more clear, we separated the figure into two Fig.S5g and Fig. S5h.

6. Fig. S6B: Two strains are labelled as FgSwp82, and none as FgSnf5, there might be a labelling problem.

Response: Sorry for the confusion. In Fig. S6b, the two strains labeled as FgSwp82 indicated 3 and 5 days of incubation, respectively. Whereas the data for Δ FgSnf5 was presented in Fig. 3f.

7. Text lines 285-288 and Fig. S9E: The peaks for the NOR gene regions (FGSG_10200 and FGSG_11585) do not look smaller or otherwise different from the wild type, contrary to what is

described in the text.

Response: You are correct. We have modified the corresponding text. See Line 308-310.

8. Fig. S9C: How was correlation calculated?

Response: Correlations between all samples were calculated by deeptools (Ramírez et al. 2016) according to raw bam files in 10 kb genomic bins. Details were provided in Figure legend. See Line 1499-1500.

Reference:

Ramírez, F., Ryan, D. P., Grüning, B., Bhardwaj, V., Kilpert, F., Richter, A. S., Heyne, S., Dündar, F., & Manke, T. (2016). deepTools2: a next generation web server for deep-sequencing data analysis. *Nucleic acids research*, 44 (W1), W160-W165.

9. Fig. S9D: The letters are much too small to be read.

Response: Thanks for pointing this out. Corrected as suggested.

10. Fig. 4F: What about the opposite, i.e. peaks that are more accessible in the two mutants compared to the wild type? Are there any, and if so, is there overlap between the mutants?

Response: Thanks for your insightful suggestion. We further analyzed accessible peaks in the Δ FgAreB and Δ FgSnf5 mutants as compared to the wild type. There are 1494 and 2063 peaks that show increased accessibility in Δ FgAreB and Δ FgSnf5 mutants, respectively (Fig. 4f, right panel). Among them, 802 peaks are co-regulated by FgAreB and FgSnf5, which accounts for 38.9% of the total increased accessible peaks in the Δ FgSnf5 mutant (Fig. 4f, right panel). Whereas, in Δ FgAreB and Δ FgSnf5, more than 83% of the inaccessible peaks in Δ FgSnf5 also become inaccessible in Δ FgAreB (Fig. 4f, left panel). These results implied that FgAreB cooperates with FgSnf5 to mainly modulate chromatin accessibility, although they also play a role in modulating chromatin inaccessibility.

After analyzed the accessible peaks in the mutants, we found that FgAreB regulated both chromatin accessibility (2050/3544 peaks) and inaccessibility (1494/3544 peaks). Similar results were reported for other pioneer TFs. For example, in human cells, the number of accessible loci modulated by TFs (the pioneer TF GATA3; the nuclear factors IB and IX, and) is comparable to that of inaccessible loci, indicating other factors are involved in accessing these loci (Adam, et al. 2020; Tanaka, et al. 2020). We added these results in the Fig. 4f and changed text. See Line 308-325.

Reference:

1. Adam, R.C., Yang, H., Ge, Y. et al. NFI transcription factors provide chromatin access to maintain stem cell identity while preventing unintended lineage fate choices. *Nat. Cell Biol.* 22, 640-650 (2020).
2. Tanaka, H., Takizawa, Y., Takaku, M. et al. Interaction of the pioneer transcription factor

GATA3 with nucleosomes. *Nat. Commun.* 11, 4136 (2020).

11. Text line 294 and Fig. 5A: It is stated that the FgIxr1 mutant shows decreased sensitivity to SNP, but in the pictures only a comparison to wild type is shown, compared to which the mutant has increased sensitivity.

Response: Thanks. We have revised Fig. 5a and added pictures taken after 3 days of inoculation. We also calculated the growth inhibition by comparing the colony diameter grown on MM-N with that on MM-N containing 10 mM SNP of each strain. The results showed that the growth inhibition rate of 10 mM SNP for PH-1 is about 30%, which is much higher than that of Δ FgIxr1. Hence, we concluded that FgIxr1 mutant showed increased resistance SNP compared with the wild type. See Fig. 5a.

12. The text and Figures describe yeast-three-hybrid assays, but these are not described in the Materials and Methods sections.

Response: Thanks. We added details describing yeast-three-hybrid assays in the Materials and Methods section as suggested. See Line 632-637.

13. Lines 364 ff.: Please describe in more detail the function of the FgTri1 protein and the text for toxisomes that was performed.

Response: Added in the results and Materials and Methods sections. See Line 402-407 and Line 581-585.

14. Line 387: The statement that this study demonstrates that FgAreB binds nucleosomal DNA might be too strong. The actual binding assays (EMSAs) were done in vitro with non-nucleosomal DNA. For the ChIP assays, it is not quite clear from the descriptions of the experiments if the status of the DNA (which would be depending on growth conditions etc.) corresponds to the one for the experiments shown e.g. in Fig. 2H.

Response: We rephrased the text as FgAreB is enriched at the corresponding region that wrapped nucleosomal DNA. We also described detailed conditions for the MNase-qPCR and ChIP-qPCR assays. See Line 206-208 and Line 210-214.

15. The RNA-seq and ATAC-seq data generated in this study should be submitted to a public database (e.g. NCBI GEO or SRA) and accession numbers need to be included in the manuscript.

Response: The corresponding RNA-seq and ATAC-seq data were deposited in NCBI (SRA), and the Bioproject number is PRJNA648257.

Reviewer #3 (Remarks to the Author):

Referee opinion on “Interplay of two transcription factors for recruitment of the chromatin

remodeling complex modulates fungal nitrosative stress response” by Jian et al.

The work is potentially of high interest but most of the conclusions drawn by the authors are not based on experimental evidence or on experiments lacking appropriate controls. Generally, a large amount of data does not replace the requirement for high quality of data.

Specific comments:

1. All experiments are performed under nitrogen starvation conditions only and no data are available under normal metabolic conditions – the explanation somewhere in the text is that “nitrogen sources lead to NO accumulation”. In this case all the effects should be even more pronounced. Clearly, the influence of a number of different nitrogen sources on NO generation and the function of the TFs and SWI/SNF needs to be tested (wild type and mutants responses).

Response: Thanks. We assessed the influence of different nitrogen sources on NO generation (Fig. S1a), and the results showed that nitrogen sources can induce NO production, which is consistent with previous reported in *Aspergillus nidulans*, *Ganoderma lucidum*, and *Magnaporthe oryzae* (Marroquinguzman et al. 2017; Zhao et al. 2020; Zhu et al. 2019). As suggested, we tested the sensitivity of the wild type PH-1 and different mutants to different nitrogen sources (see Revised Fig. S2a). $\Delta FgAreB$, $\Delta FgSnf5$ and triple mutants were sensitive to the tested nitrogen sources.

Reference:

1. Marroquin-guzman M, Hartline D, Wright JD, Elowsky C, Bourret TJ, Wilson RA (2017) The *Magnaporthe oryzae* nitrooxidative stress response suppresses rice innate immunity during blast disease. *Nature Microbiol* 2: 17054.
2. Zhao Y, Lim J, Xu J, Yu J, Zheng W (2020) Nitric oxide as a developmental and metabolic signal in filamentous fungi. *Molecular Microbiology* 113(5):872-882.
3. Zhu J, Sun Z, Shi D, Song S, Lian L, Shi L, Ren A, Yu H, Zhao M (2019) Dual functions of AreA, a GATA transcription factor, on influencing ganoderic acid biosynthesis in *Ganoderma lucidum*. *Environmental Microbiology* 21: 4166-4179.

2. The authors screened 1500 deletions mutants? No description of the screen or the mutant library is provided

Response: The mutant library in our lab includes mutants of 71 phosphatase genes, 94 kinase genes, 180 transcription factor genes, 60 ABC transporter genes, 58 MFS transporter genes, 84 secondary metabolism cluster genes, 26 RNA binding protein genes, 14 ergosterol biosynthesis genes, 118 epigenetic factor genes, 12 histone deacetylase-encoding genes, 28 acetyltransferase-encoding genes, 23 genes involved in SUMOylation, 85 potential NLR genes, 12 secreted protein genes, 21 ubiquitin related genes, 185 enzymes involved in amino acid, ROS, DON, ZEN, phospholipid biosynthesis/metabolism and redox reaction, 263 response genes to fungicides or chemicals (including phenamacril, tebuconazole, prothioconazole, rapamycin, pydiflumetofen and DNA damage agents) and other functional genes participating in fungal growth and development.

Some genes have been published in previous studies, including:

1. Yun, Y., Liu, Z., Yin, Y., Jiang, J., Chen, Y., & Jin - Rong Xu, et al. (2015) Functional analysis of the *Fusarium graminearum* phosphatome. *New Phytologist*, 207(1).
2. Wang, C., Zhang, S., Hou, R., Zhao, Z., & Xu, J.R. (2010) Functional analysis of the kinome of the wheat scab fungus *Fusarium graminearum*. *Plos Pathogens*, 7(12), e1002460.
3. Yin Y, Wang Z, Cheng D, et al. (2018) The ATP-binding protein FgArb1 is essential for penetration, infectious and normal growth of *Fusarium graminearum*. *New Phytologist*. 219(4):1447-1466.
4. Chen Y, Wang J, Yang N, Wen Z, Sun X, Chai Y, Ma Z. (2018) Wheat microbiome bacteria can reduce virulence of a plant pathogenic fungus by altering histone acetylation. *Nat Commun*. 24;9(1):3429.
5. Liu Z, Jian Y, Chen Y, Kistler HC, He P, Ma Z, Yin Y. (2019) A phosphorylated transcription factor regulates sterol biosynthesis in *Fusarium graminearum*. *Nat. Commun.* **10**, 1228.

3. Authors state that it is FgAreB that binds to NS promoters – controls of mutants in other GATA-factor genes (AreA, SreA, etc.) are missing. No conclusions can be drawn from the present EMSAs (from line 176 onwards).

Response: Thanks! We tested the sensitivity of other GATA TF mutants including FgSreA and FgAreA mutants to SNP in both MM-N and PDA media and added the data in revised Fig. S2c. Δ FgAreA is not able to grow on MM-N, which is consistent with previous report that FgAreA positively regulates nitrogen metabolism and the deletion mutant of FgAreA is not able to grow on medium with secondary nitrogen sources (Giese et al. 2013). In contrary to Δ FgAreB, both FgAreA and FgSreA mutants did not increase sensitivity to 2 mM SNP (Fig. S2c ; Fig. 1d).

In addition, we purified the GATA domain of FgAreA and FgSreA fused His-tag proteins and EMSA assays showed that FgAreA and FgSreA do not bind directly to NS promoters. See revised Fig. S4c and S4d.

Reference:

1. Giese H, Sondergaard TE, Sorensen JL. 2013. The AreA transcription factor in *Fusarium graminearum* regulates the use of some non-preferred nitrogen sources and secondary metabolite production. *Fungal Biol* 117: 814-821.

4. It is clear that AreB seems to be involved – but if it is the “pioneer TF” initializing PIC assembly and SWI/SNF recruitment remains enigmatic from the experimental data. In spite of this, the authors develop a wild hypothesis on NS-gene regulation by AreB and competition by Ixr1 that is totally unjustified. No direct in vivo competition experiments (e.g. sequential ChIP-on-ChIP) showing competition are provided.

Response: Pioneer factors can function actively by helping to open or organize the local chromatin, which in turn allows the binding of other transcription factors, chromatin modifiers, and

nucleosome remodelers (Zaret et al. 2011). Pioneer factors bind to the condensed chromatin and then recruit chromatin modifiers like SWI/SNF complex to help open the local chromatin and facilitate gene transcription (Iwafuchi-Doi M & Zaret KS, 2014; Wang et al. 2014; Zaret et al. 2016). These observations suggest that both pioneer factors and chromatin modifiers are required for initializing gene transcription. We showed that FgAreB has the ability to bind the target sequence in absence of nitrosative stress stimuli (Fig. 2g); and nitrosative stress enhances the interaction between FgAreB and SWI/SNF complex (Fig. 3e). ChIP-qPCR assays showed that enrichment of FgAreB-GFP at the promoters of NS response genes was independent of FgSnf5 (Fig. 4b), but FgSnf5 was dependent on FgAreB (Fig. 4a), implying that FgAreB recruits the SWI/SNF complex to the promoters of NS response genes. Taken together, our data supported that FgAreB is required for the enrichment of SWI/SNF complex at promoter regions of NS response genes. However, our current data don't support the function of FgAreB in initializing PIC assembly.

In addition, we showed that the GATA domain of FgAreB is critical for its interaction with FgIxr1 or SWI/SNF complex. Moreover, NS promotes the degradation of FgIxr1 (Fig. 5i-k), but not that of FgSnf5 (Fig. S10j). Combining the results of yeast three hybrid and GST pull down assays (Fig. 5g and h), we concluded that FgIxr1 competes with the SWI/SNF complex for binding to GATA domain of FgAreB, thereby attenuating the interaction between FgAreB and SWI/SNF complex. Since these were *in vitro* data, we thus conducted the Co-IP assays to confirm that FgIxr1 competes with SWI/SNF complex for FgAreB *in vivo*. Deletion of FgIxr1 increased the interaction of FgAreB and FgSwp73 (see revised Fig. S10g). Overall, our data indicate that FgIxr1 competes with SWI/SNF to interact with FgAreB, and subsequently decreases transcription of NS response genes.

Reference:

1. Zaret, K. S., & Carroll, J. S. (2011). Pioneer transcription factors: establishing competence for gene expression. *Genes & development*, 25(21), 2227-2241.
2. Iwafuchi-Doi M, Zaret KS. (2014) Pioneer transcription factors in cell reprogramming. *Genes & Development*. 28(24):2679-2692.
3. Wang B, Kettenbach AN, Gerber SA, Loros JJ, Dunlap JC. (2014) *Neurospora* WC-1 Recruits SWI/SNF to Remodel frequency and Initiate a Circadian Cycle. *PLoS Genet* 10(9): e1004599.
4. Zaret, Kenneth S. and Mango, Susan E. (2016) Pioneer transcription factors, chromatin dynamics, and cell fate control. *Current opinion in genetics & development*, 37. 76-81.

5. Line 64-Intro: it is not true that TF have an initializing role in binding to closed chromatin – in contrary, they usually bind to NFRs and then recruit what it takes to open/remodel chromatin.

Response: General TFs indeed don't have the abilities to bind closed chromatin, but "pioneer" factors have been found to be able to access their target DNAs wrapping nucleosomes. Some pioneer factors can function actively by helping to open or organize the local chromatin, which in

turn allow the binding of other transcription factors, chromatin modifiers, and nucleosome remodelers to special chromatin regions (Zaret et al. 2011; Iwafuchi-Doi M & Zaret KS, 2014; Zaret et al. 2016). As an example, pioneer factors FoxA and GATA are initial and important chromatin-binding factors in plants (Zaret et al. 2011; Albergaria et al. 2009; Cirillo et al. 2002; Takaku et al. 2016). In addition, GATA3 in mammalian, which is homolog of FgAreB, is a pioneer transcription factor and is able to bind the closed chromatin (Tanaka et al. 2020). Based on these previous reports, pioneer transcription factors may possess the ability to bind closed chromatin region.

Reference:

1. Cirillo L, Lin FR, Cuesta I, Jarnik M, Friedman D, Zaret K. 2002. Opening of compacted chromatin by early developmental transcription factors HNF3 (FOXA) and GATA-4. *Mol Cell* 9: 279-289.
2. Albergaria A, Paredes J, Sousa B, Milanezi F, Carneiro V, Bastos J, Costa S, Vieira D, Lopes N, Lam EW, et al. 2009. Expression of FOXA1 and GATA-3 in breast cancer: the prognostic significance in hormone receptor-negative tumours. *Breast Cancer Res* 11: R40.
3. Zaret, K. S., & Carroll, J. S. (2011). Pioneer transcription factors: establishing competence for gene expression. *Genes & development*, 25(21), 2227-2241.
4. Iwafuchi-Doi M, Zaret KS. (2014) Pioneer transcription factors in cell reprogramming. *Genes & Development*. 28(24):2679-2692.
5. Takaku, M., Grimm, S.A., Shimbo, T., Perera, L., Menafra, R., Stunnenberg, H.G., Archer, T.K., Machida, S., Kurumizaka, H. and Wade, P.A., 2016. GATA3-dependent cellular reprogramming requires activation-domain dependent recruitment of a chromatin remodeler. *Genome biology*, 17(1), pp.1-16.
6. Zaret, Kenneth S. and Mango, Susan E. (2016) Pioneer transcription factors, chromatin dynamics, and cell fate control. *Current opinion in genetics & development*, 37. 76-81.
7. Tanaka, H., Takizawa, Y., Takaku, M., Kato, D., Kumagawa, Y., Grimm, S.A., Wade, P.A. and Kurumizaka, H., 2020. Interaction of the pioneer transcription factor GATA3 with nucleosomes. *Nature communications*, 11(1), pp.1-10.

6. Lines 100-102: solvent controls are missing.

Response: Thanks. The solvent for cPTIO is ddH₂O and DAF-FM DA is PBS. Plant tissues were stained by DAF-FM DA without (H₂O) or with the NO scavenger cPTIO, therefore, as we performed the staining assays, both of the solvent were added to the sample. We included the solvent control in revised Fig. 1a and Figure legends section, Line 1056-1059.

7. Line 114: growth test under these conditions not valid – nitrogen supply also needed and biomass measurements; colony extension does not represent biomass in fungi grown on solid media.

Response: We added nitrogen supply tests for each mutant and presented the results in Revised Fig. S2a. In addition, we also tested SNP sensitivity on MM-N medium because nitrogen sources can induce NO production (Marroquin-guzman et al. 2017; Zhao et al. 2020; Zhu et al. 2019) (Fig. S1a). Since mycelia grown on MM-N media are sparse (Fig. 1c), it is extremely difficult to measure biomass on MM-N without or with SNP. Therefore, we decided to test SNP sensitivity of the wild type strain PH-1 on PDA and MM without or with SNP on solid media and liquid PDB and YEPD (see revised Fig. S1d and e). SNP at 10 mM significantly inhibited mycelia growth on both PDA and MM and decreased fungal biomass in PDB and YEPD, which is consistent with the phenotype on plate.

Reference:

1. Marroquin-guzman M, Hartline D, Wright JD, Elowsky C, Bourret TJ, Wilson RA (2017) The *Magnaporthe oryzae* nitrooxidative stress response suppresses rice innate immunity during blast disease. *Nature Microbiol* 2: 17054.
2. Zhao Y, Lim J, Xu J, Yu J, Zheng W (2020) Nitric oxide as a developmental and metabolic signal in filamentous fungi. *Molecular Microbiol* 113(5):872-882.
3. Zhu J, Sun Z, Shi D, Song S, Lian L, Shi L, Ren A, Yu H, Zhao M (2019) Dual functions of AreA, a GATA transcription factor, on influencing ganoderic acid biosynthesis in *Ganoderma lucidum*. *Environmental Microbiol* 21: 4166-4179.

8. Line 137: in many case the GATA domain overlaps with the NLS – is this the case also here? A putative NLS search result should be included

Response: Thanks. We predicted the GATA domain and NLS of GATA transcription factors, FgAreA, FgSreA, FgAreB, and MoAsd4 (the homolog of FgAreB in *Magnaporthe oryzae*) and data are shown in the following Figure. The NLS signal of FgSreA shares partial overlap with one of its GATA domain, but the predicted NLSs of FgAreA, FgAreB and MoAsd4 are outside of their GATA domains, which is consistent with our results that deletion of GATA domain of FgAreB didn't change the nuclear localization of FgAreB (Fig. S3h).

9. Line 167: statement needs triple mutant addition to experimental design

Response: Previous studies showed that five groups of NS response genes are responsible for detoxifying NO in fungi, i.e. flavohaemoglobins (FHB), S-nitrosoglutathione reductase (GSNOR),

P450 nitric oxide reductases (NOR), porphobilinogen deaminase (HEMC), and nitrosothionein (Canovas et al. 2016; Zhao et al. 2020; Zhou et al. 2013). BLASTp search showed that *Fg* harbors five putative NS response genes, e.g. *FgFHB1*, *FgFHB2*, *FgGSNOR*, *FgNOR* and *FgHEMC*, that belong to four groups of NS response genes, where *FgFHB1* and *FgFHB2* belong to flavohaemoglobins (FHB) group. We speculated that disruption of one or two genes is not sufficient to impair the entire NO detoxification pathway. Therefore, we generated triple mutants of NS response genes that are responsible for detoxifying NO using the homology recombinant strategy with three different selection markers (*NTC*: Nourseothricin resistance cassette; *HPH*: hygromycin resistance cassette; *G418*: Geneticin resistance cassette). We described the details for constructing triple mutants in Revised Figure S4a. Due to the lack of a fourth selectable marker in *Fg*, we were unable to generate a four-gene mutant, although we believe that the NO detoxification pathway would be more impaired in four-gene mutants.

Reference:

1. Canovas, D., Marcos, J. F., Marcos, A. T., & Strauss, J. (2016) Nitric oxide in fungi: is there NO light at the end of the tunnel? *Current Genetics*, **62**(3): 513-518.
2. Zhao Y, Lim J, Xu J, Yu J, Zheng W (2020) Nitric oxide as a developmental and metabolic signal in filamentous fungi. *Molecular Microbiology* **113**(5):872-882.
3. Zhou S, Narukami T, Masuo S, Shimizu M, Fujita T, Doi Y, Kamimura Y, Takaya N (2013) NO-inducible nitrosothionein mediates NO removal in tandem with thioredoxin. *Nature Chemical Biology* **9**: 657-663.

10. Fig 2(lines ~190): controls of other GATA factors missing, control without SNP missing, - statement invalid, all the effects could be indirect

Response: We added *FgAreA* and *FgSreA* as controls; and found that *FgAreA* and *FgSreA* were not enriched at *FgGSNOR* promoter under both non-nitrosative and nitrosative stress conditions (Fig. S4e). Also see comments above.

11. Co-IP and Y2H data contradictory for Snf5 and Snf2

Response: Y2H assays reveal direct interaction between two tested proteins, while Co-IP assays can also provide information about indirect interaction among proteins coexisting in a higher order protein complex.

In this study, Y2H data in Fig. 3a and Fig. S5a showed *FgAreB* could only directly interact with *FgSwp73* that is consistent with GST pull down assay, but not directly interact with *FgSnf2* and *FgSnf5*. Thus, we performed Co-IP to test whether *FgAreB* could indirectly associate with *FgSnf2* and *FgSnf5* *in vivo*. As shown in Fig. S5g and h, *FgAreB* is associated with *FgSnf2* and *FgSnf5*, and their associations are enhanced by SNP treatment. Since *FgSwp73* directly interacts with *FgSnf2* and *FgSnf5* in Y2H assays (Fig. 3b), it is reasonable for us to speculate that *FgSwp73* may serve as a bridge to link the interaction between *FgAreB* and *FgSnf2* or *FgSnf5* indirectly.

Reference:

1. Struk, S., Jacobs, A., Sánchez Martín-Fontecha, E., Gevaert, K., Cubas, P., & Goormachtig, S. (2019). Exploring the protein-protein interaction landscape in plants. *Plant, cell and environment*, 42, 387-409.
2. Xing, S., Wallmeroth, N., Berendzen, K.W. and Grefen, C., 2016. Techniques for the analysis of protein-protein interactions *in vivo*. *Plant physiology*, 171(2), pp.727-758.

12. Line 255: areB presence is not SNP dependent – this is a contradiction to statements above

Response: Yes, we showed that FgAreB presence is not dependent on SNP and FgSnf5 (Fig. 2g, Fig. 4b). NS response genes were activated under SNP treatment condition (Fig. 2a), however, FgAreB-GFP was still able to enrich at the promoters of four NS response genes under non-SNP treatment in the ChIP-qPCR assays (Fig. 2g). This result is consistent with the key features of pioneer factors as reviewed by Zaret et al (See comments for question #5 above). Since we only described the enrichment of FgAreB in the NS gene promoter under 10 mM SNP (Fig. 2g), in order to avoid confusion, we modified the corresponding text. See Line 205-214.

Reference:

Zaret, K. S., & Carroll, J. S. (2011). Pioneer transcription factors: establishing competence for gene expression. *Genes & development*, 25(21), 2227-2241.

13. Line 280: MNase assays as done here are not quantitative – statement needs to be removed or experimental design needs to be changed

Response: As suggested, we changed the experimental design. See revised Fig. 4d. In the original Fig. 4d, the MNase-qPCR assays were carried out under SNP treatment in PH-1, Δ FgAreB and Δ FgSnf5. By comparing to Fig. 2h, where the MNase-qPCR assay was conducted under non-nitrosative stress condition, the nucleosome occupancy significantly reduced under SNP-treated condition (Fig. 4d). We therefore assessed the nucleosome occupancy around the FgAreB putative binding site of *FgGSNOR* promoter in the PH-1, Δ FgAreB and Δ FgSnf5 mutants with or without SNP treatment. We showed that the nucleosome occupancy at the *cis* element was significantly reduced in PH-1 after SNP treatment, whereas it was much higher in Δ FgAreB and Δ FgSnf5 mutants regardless the conditions (Revised Fig. 4d). These results indicate that loss of FgAreB or FgSnf5 affect nucleosomes retention at *FgGSNOR* promoter under SNP treatment.

14. Line 267: conditions for RNA seq are not described

Response: Thanks. Added conditions for RNA-seq in the revised text. See Line 1211-1212.

15. Line 298: Ixr1 tagging is not described

Response: The FgIxr1 tagging information was shown in Figure legend Fig. S10b, Line 1512 and

we also added the details in the results part, Line 334.

16. Line 304: conditions are not described for nuclear positioning observations.

Response: The condition for nuclear positioning observations was shown in Figure legend Fig. 5c, Line 1241-1242. We also described in the results part, See Line 340-341.

17. Line 308: statement invalid: it could be other GATA factors that interact with their GATA domains – shown that this domain is crucial for interaction – so it could easily be other factors that pull this TF into the nucleus

Response: In Fig. 5d and e, the truncated FgAreB without the GATA domain (FgAreB^{ΔGATA}) significantly decreased its interaction with FgIxr1 based on Y2H and Co-IP assays. In Y2H assays, FgAreB and FgAreB^{ΔGATA} were fused with GAL4 activation domain (AD) of pGADT7 vector and the fusion protein is targeted to the yeast nucleus by the SV40 nuclear localization signal (NLS). Thus, the FgAreB^{ΔGATA} can constitutively present in nucleus in Y2H assays while we cannot rule out that other factors interacting with FgAreB GATA domain to pull FgAreB into nucleus. In addition, as shown in Fig. S3h, FgAreB^{ΔGATA}-GFP presented in nucleus as FgAreB-GFP in *F. graminearum* hyphae. These results indicate that decreased interaction of FgAreB^{ΔGATA} with FgIxr1 is due to deletion of the FgAreB GATA domain.

pGADT7 Vector Information (Clontech)

18. Fig 5/lane 330: how significant are these statements based fluorescence measurements? What is the statistics, which hyphae are considered?

Response: The fluorescence pictures of Fig. 5j and k were representative hyphae under *in vitro* and *in vivo* nitrosative stress conditions, respectively. The panels under fluorescence pictures were western blot results, which showed protein intensity of FgIxr1 under corresponding *in vitro* and *in vivo* nitrosative stress conditions. The protein intensity was quantitatively analyzed as indicated, which represented the total protein of the treated hyphae, not a specific hypha.

19. Discussion lines 485ff: I wonder why in all these interaction screens the known TFs ANK1 and ZC1 are not appearing? This should be discussed.

Response: A recent study showed that FgANK1 is an Ankyrin-Repeat containing protein and regulates NO biosynthesis of *Fg* during the pre-contact stage with host root. In the absence of host

signal, FgANK1 resides in the cytoplasm, while FgANK1 translocates into the nucleus and interacts with zinc finger transcription factor FgZC1 in response to host signals. The FgANK1 and FgZC1 complex directly bind to the nitrate reductase (NR) promoter to regulate host root-trigger NO production. These data suggest that the **FgANK1 and FgZC1 mainly regulated NO production** and the corresponding mutants lose the ability to produce NO. This function of FgANK1-ZC1 is different from that of FgAreB. In this study, **FgAreB and FgSnf5 were found to modulate NO detoxification** via binding NO detoxification gene promoters. It is possible that the signal pathway for NO biosynthesis may differ from that of NO detoxification, therefore, it is understandable that we did not identify FgANK1 and FgZC1 in our yeast two hybrid screening. We added this point in discussion, see Line 518-526.

Reference:

Ding, Y., Gardiner, D.M., Xiao, D. and Kazan, K., (2020). Regulators of nitric oxide signaling triggered by host perception in a plant pathogen. *Proceedings of the National Academy of Sciences*, 117(20), pp.11147-11157.

20. Methods: how were the ChIP antibodies validated? Solvent control for chemo-biological experiments generally missing

Response: The ChIP grade antibodies used in this study were validated by western blot assays to test specificity to proteins (see below). The commercial antibodies used for ChIP-qPCR assays were directly applied to each sample, we didn't use any other solvent. In addition, IgG is used as the control for ChIP-qPCR assays instead, which is a common negative control antibody to perform ChIP-qPCR assays (Liu et al. 2017; Cheng et al. 2018; Liu et al. 2019; Sandmann et al. 2006). We added the corresponding information in Materials and Methods of the revised manuscript. See Line 602-604.

Reference:

1. Sandmann, T., Jakobsen, J. S., & Furlong, E. E. (2006). ChIP-on-chip protocol for genome-wide analysis of transcription factor binding in *Drosophila melanogaster* embryos. *Nature protocols*. 1(6), 2839.
2. Liu X, Dang Y, Matsu-Ura T, He Y, He Q, Hong CI, Liu Y. (2017) DNA replication is required for circadian clock function by regulating rhythmic nucleosome composition. *Mol Cell*. Jul

20;67(2):203-213.e4.

3. Cheng S, Tan F, Lu Y, Liu X, Li T, Yuan W, Zhao Y, Zhou DX. (2018) WOX11 recruits a histone H3K27me3 demethylase to promote gene expression during shoot development in rice. *Nucleic Acids Res.* 46(5):2356-2369.

4. Liu Z, Jian Y, Chen Y, Kistler HC, He P, Ma Z, Yin Y. (2019) A phosphorylated transcription factor regulates sterol biosynthesis in *Fusarium graminearum*. *Nat. Commun.* **10**, 1228.

REVIEWERS' COMMENTS

Reviewer #2 (Remarks to the Author):

The concerns that I had about the manuscript have been addressed.

Reviewer #3 (Remarks to the Author):

In the revised version the authors picked up basically all comments and sincerely dealt with the criticisms. All crucial controls were added and even a whole experimental approach (chromatin accessibility) was changed due to the criticism. The paper is now very good and complete. It provides a novel and now scientifically sound insight into the NO response of *F. graminearum*. Specific points that were changed according to suggestions or criticism:

- 1&7. effect of different nitrogen sources on NO generation was assessed now. Please correct reference for *A. nidulans* from Zhao et al., to Marcos et al. 2016, Mol.Microbiol.
- 2. mutants contained in the library have been listed now in the rebuttal letter - this would be a valuable information for the Fusarium community and should be added as supplementary table to the paper
- 3&10. sensitivity of AreA and SreA mutants were tested
- 4&5. a description on what "pioneer" TFs do has been added now in the discussion and the Ixr1 competition statement has been backed up with more experimental data.
- 6. controls for solvents have been included
- 8. the NLS data were added
- 9. triple mutant studies were added
- 13. MNase accessibility: the experimental design was changed so that quantitative analysis is possible
- 14&15. conditions for RNAseq and Ixr1 tagging are described now
- 19: the role of NO production by the recently identified ANK1/ZC1 TFs is discussed
- 20: a Western showing antibody validation was added to supplementary figures

Responses to reviews:

Reviewer #3 (Remarks to the Author):

In the revised version the authors picked up basically all comments and sincerely dealt with the criticisms. All crucial controls were added and even a whole experimental approach (chromatin accessibility) was changed due to the criticism. The paper is now very good and complete. It provides a novel and now scientifically sound insight into the NO response of *F. graminearum*.

Thanks a lot.

Specific points that were changed according to suggestions or criticism:

1&7. effect of different nitrogen sources on NO generation was assessed now. Please correct reference for *A. nidulans* from Zhao et al., to Marcos et al. 2016, Mol.Microbiol.

Response: We corrected the reference as suggested.

2. The mutants contained in the library have been listed now in the rebuttal letter - this would be a valuable information for the *Fusarium* community and should be added as supplementary table to the paper

Response: We added the library of mutants in the Supplementary Information 1.

3&10. sensitivity of AreA and SreA mutants were tested

4&5. a description on what "pioneer" TFs do has been added now in the discussion and the Ixr1 competition statement has been backed up with more experimental data.

6. controls for solvents have been included

8. the NLS data were added

9. triple mutant studies were added

13. MNase accessibility: the experimental design was changed so that quantitative analysis is possible

14&15. conditions for RNAseq and Ixr1 tagging are described now

19: the role of NO production by the recently identified ANK1/ZC1 TFs is discussed

20: a Western showing antibody validation was added to supplementary figures

Thank you very much for pointing out the changes.